# Query Circuits: Explaining How Language Models Answer User Prompts

## Abstract

Explaining why a language model produces a particular output requires local, input-level explanations. Existing methods uncover global capability circuits (e.g., indirect object identification), but not why the model answers a specific input query in a particular way. We introduce *query circuits*, which directly trace the information flow inside a model that maps a specific input to the output. Unlike surrogate-based approaches (e.g., sparse autoencoders), query circuits are identified within the model itself, resulting in more faithful and computationally accessible explanations. To make query circuits practical, we address two challenges. First, we introduce Normalized Deviation Faithfulness (NDF), a robust metric to evaluate how well a discovered circuit recovers the model's decision for a specific input, and is broadly applicable to circuit discovery beyond our setting. Second, we develop sampling-based methods to efficiently identify circuits that are sparse yet faithfully describe the model's behavior. Across benchmarks (IOI, arithmetic, MMLU, and ARC), we find that there exist sparse query circuits within the model that recover much of its performance on single queries. For example, on average, a circuit covering only 1.3% of model connections can recover about 60% of performance on an MMLU question. Overall, query circuits provide a step towards faithful, scalable explanations of how language models process individual inputs.

## 1 Introduction

Explaining the decisions of large language models (LLMs) is essential for their deployment in high-stakes domains such as medicine (Amann et al., 2020) and autonomous systems (Omeiza et al., 2022). For example, when a medical AI agent receives a query from clinicians and decides whether a patient should undergo surgery, its reasoning must be interpretable to ensure the decision does not rely on spurious/shortcut features (Yuan et al., 2024); when an autonomous vehicle selects an incorrect control action, its failure mode must be explainable to allow accurate attribution of responsibility.

Recently, circuit discovery (Conmy et al., 2023; Hanna et al., 2024; Ameisen et al., 2025) has emerged as a popular approach for explaining model mechanisms (Kharlapenko et al., 2025). However, most studies only investigate circuits of simple inference patterns, such as indirect object identification (IOI) (Wang et al., 2023) and greater-than (GT) comparison (Hanna et al., 2023). Though valuable, these circuits do not explain how a model produces a particular output for a given user input query.

Other approaches identify circuits in surrogate models such as sparse autoencoders (SAEs) (Huben et al., 2024) and cross-layer transcoders (CLTs) (Lindsey et al., 2024). Circuit discovery is easier in surrogates due to their sparse activations. Recently, a CLT-based prompt-level circuit discovery framework, called Circuit Tracing (Ameisen et al., 2025), is proposed to interpret input-specific model behavior. However, surrogate models often fail to faithfully reconstruct model activations (Ameisen et al., 2025) and may not capture the true mechanisms of the LLM (Marks et al., 2024; Olah, 2025), undermining their reliability. Moreover, training surrogates is computationally expensive (Templeton et al., 2024), limiting their accessibility. Other prior studies, such as circuit discovery in vision models (Kwon et al., 2025) and input-dependent feature analyses in LLMs (Chen et al., 2024; Ghandeharioun et al., 2024), likewise do not provide prompt-level explanations without relying on surrogate models.

We introduce the task of **query circuit discovery**: uncovering the specific circuit *directly inside (i.e., in-place)* an LLM that drives its decision on a single input query. Prior work has uncovered in-place *capability circuits*—sub-networks that implement global skills such as indirect object

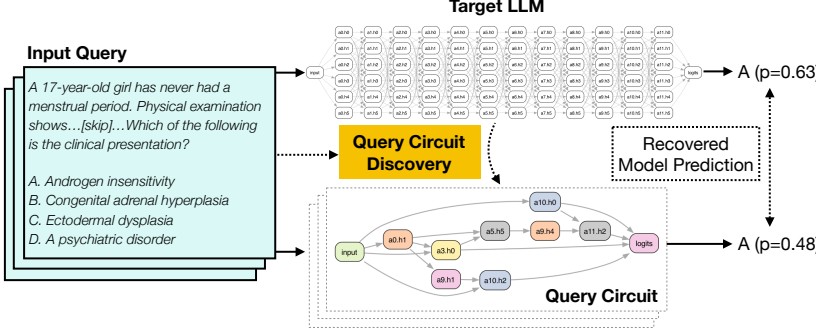

Figure 1: **Query circuit discovery aims to identify a sparse sub-network within the LLM that underlies the model response to a user input query.** The LLM and circuit in this illustration are simplified for visualization.

identification—but these do not explain why the LLM produces a particular output for a given input prompt. Instead, query circuits provide local, prompt-level explanations by directly tracing the information flow inside the LLM (Figure 1).

We highlight key technical challenges of query circuit discovery and propose methods to address them. First, the widely adopted Normalized Faithfulness Score (NFS) (Hanna et al., 2024; Zhang et al., 2025; Marks et al., 2025) used to assess how well a circuit recovers the model performance becomes unstable on general datasets (e.g., MMLU), and thus fails to reliably indicate when circuits of increasing size begin to capture model behavior. We therefore introduce *Normalized Deviation Faithfulness (NDF)*, a more robust metric for query circuit evaluation. Second, existing methods from capability circuit discovery often fail to yield compact and faithful query circuits. To overcome this, we propose to use *Best-of-N (BoN)* sampling and two variants—interpolated BoN (iBoN) and BoN with constraint-adaptive score matrix (BoN-CSM)—which reliably recover faithful query circuits.

Experimental results across multiple benchmarks (IOI, arithmetic, MMLU (Hendrycks et al., 2021), and ARC Challenge (Clark et al., 2018)) show that *even for complex natural queries, compact query circuits can still be found within the LLM that account for a considerable portion of its responses*. For example, using BoN, we find that for a multiple-choice question (MCQ) in MMLU, a query circuit with only 1.3% of the target LLM's edges can, on average, recover roughly 60% of the model's behavior on that query. In summary, our contributions are threefold:

- We formulate the task of **query circuit discovery**, contrasting it with both capability circuit discovery and surrogate-model-based approaches.
- We identify and address two key technical challenges: (i) unreliable evaluation of query circuits by the previous metric (NFS), for which we propose *Normalized Deviation Faithfulness (NDF)*; and (ii) failure of existing methods to find compact and faithful query circuits, for which we propose *Best-of-N* (BoN) sampling and its variants.
- Across diverse datasets, we demonstrate that even a small circuit within the model can explain much of the model behavior on the individual query, showing query circuit discovery as a practical path toward faithful and scalable prompt-level LLM decision explanations.

## 2 BACKGROUND: CAPABILITY CIRCUIT DISCOVERY

This section provides the technical background of circuit discovery. Section 2.1 reviews transformer circuits, while Section 2.2 introduces capability circuit discovery, including how to discover a circuit (Section 2.2.1) and how to evaluate both the circuit and the discovery method (Section 2.2.2).

### 2.1 TRANSFORMER CIRCUITS

Transformer circuits (Elhage et al., 2021) represent an LLM $M$ as a directed acyclic graph with node and edge sets $\{V, E\}$, where, following prior work (Syed et al., 2024; Conmy et al., 2023), each node in $V$ is an MLP or attention head, and edges in $E$ are where the outputs of earlier nodes feed into later ones, defined via *residual rewrite* (Elhage et al., 2021; Nanda & Bloom, 2022). WLOG, a circuit can be specified by its edge set $E$, omitting the explicit node set $V$. For a given phenomenon or capability,

a compact and faithful circuit that captures the critical information flows among components enables more precise and efficient interpretability research (Quirke & Barez, 2024; Lan et al., 2024).

## 2.2 CAPABILITY CIRCUIT DISCOVERY

Given a target LLM $M$ with edge set $E$ and a capability of interest (e.g., IOI), capability circuit discovery aims to identify a capability circuit $C_c$ with edge set $E_c \subset E$ that captures $M$'s underlying mechanisms for this capability. To study the capability, it is instantiated as a dataset $D$ of queries, where each query is designed so that answering it correctly requires the model to use that capability.

### 2.2.1 EDGE SCORING AND CIRCUIT CONSTRUCTION

Prior methods from capability circuit discovery typically construct the circuit by selecting edges based on their influence on the model's outputs. An edge $e$'s importance score $a_e$ is defined as its averaged indirect effect (IE) (Vig et al., 2020) on the model's performance over the dataset $D$:

$$a_e := \frac{1}{|D|} \sum_{q \in D} \Big( L(M(q \,|\, \mathrm{do}(e \leftarrow e'))) - L(M(q)) \Big), \tag{1}$$

where $L(\cdot)$ is a performance metric for each query $q$, such as the logit difference between the correct and incorrect tokens (Heimersheim & Nanda, 2024). The operator $\mathrm{do}(e \leftarrow e')$ denotes corrupting edge $e$ by replacing its propagated feature with a corrupted feature $e'$. The $e'$ is obtained by feeding the LLM a corrupted query $q'$, constructed by removing the key factual or linguistic cue in the original query $q$ that guides the model's solution. Details of corrupted queries for different question types we studied are provided in Appendix A. The scores of all edges can arrange as an edge score matrix $S \in \mathbb{R}^{n \times n}$, where $n$ denotes the number of nodes. Notably, Equation 1 is not additive, i.e., $a_{e_i \cup e_j} \neq a_{e_i} + a_{e_j}$, where $a_{e_i \cup e_j}$ denotes the effect of corrupting $e_i$ and $e_j$ in the same forward pass.

Approaches that compute $a_e$ directly via Equation 1, such as ACDC (Conmy et al., 2023), are referred to as edge activation patching methods (Zhang & Nanda, 2024). They require two forward passes of $M$ to score each edge. To improve efficiency, some recent studies (Hanna et al., 2024; Marks et al., 2025) reformulate IE computation as integrated gradients (IG) (Sundararajan et al., 2017):

$$a_e = (e - e')^\top \int_0^1 \nabla_e M\big(z' + \alpha(z - z')\big)\, d\alpha \approx (e - e')^\top \frac{1}{m} \sum_{k=1}^{m} \nabla_e M\big(z' + \tfrac{k}{m}(z - z')\big), \tag{2}$$

where $z$ and $z'$ are the token embeddings of $q$ and $q'$. $m$ is the discretization step. Averaging over $D$ is omitted for simplicity. Equation 2 approximates all edges' IEs in parallel, requiring a fixed number of forward passes regardless of the edge count. Approaches applying Equation 2, such as EAP (Syed et al., 2024)[1] and EAP-IG (Hanna et al., 2024), are referred to as edge attribution patching methods.

Using the computed edge scores, capability circuit discovery methods construct the capability circuit $C_c$ given a budget of $N$ edges. Two straightforward approaches are: (i) greedily selecting $N$ edges with the highest scores (Hanna et al., 2024), and (ii) selecting nodes or edges whose scores exceed a predefined threshold (Conmy et al., 2023; Marks et al., 2025). A more sophisticated method is Dijkstra-like iterative construction (Conmy et al., 2023; Hanna et al., 2024): Start from the logit node and iteratively add back influential edges whose child node is already included in the circuit.

### 2.2.2 EVALUATION OF CAPABILITY CIRCUIT AND DISCOVERY METHOD

Normalized Faithfulness Score (NFS) (Marks et al., 2025; Zhang et al., 2025; Mueller et al., 2025) has been widely adopted to quantify how well the discovered capability circuit $C_c$ recovers the original LLM $M$'s performance on $D$. It is defined as:

$$NFS(C_c) := \frac{L(C_c(D)) - L(M(D'))}{L(M(D)) - L(M(D'))}, \tag{3}$$

where $L(C_c(D))$ denotes the overall performance of $C_c$ on $D$. $D' := \{q_i' \mid q_i \in D\}$. $NFS(C_c)$ measures the fraction of $M$'s performance on $D$ recovered by $C_c$. $NFS(C_c) = 1$ indicates $C_c$

---

[1]Although the original EAP paper (Syed et al., 2024) frames it as a linear approximation of Equation 1 via the Taylor series, it can also be interpreted as applying integrated gradients with a discretization step of $m = 1$.

perfectly recovers $M$'s performance. $NFS(C_c) = 0$ means $C_c$ performs the same as $M$ on corrupted queries. In toy tasks (e.g., IOI), where capability circuits have mainly been studied, NFS typically falls within $[0, 1]$ (Hanna et al., 2024), although its definition (Equation 3) does not guarantee boundedness. A discovery method is more effective if, across varying numbers of edges $N$, it consistently identifies circuits with higher (close to 1) NFS than the counterparts (i.e., a better Pareto frontier).

## 3 PROBLEM FORMULATION: QUERY CIRCUIT DISCOVERY

### 3.1 OBJECTIVE

Our proposed query circuit discovery seeks methods that, for any natural query $q$ and an edge budget $N$, consistently identify a faithful circuit $C_q$ defined by the edge set $E_q \subset E$ that captures the mechanisms by which the target LLM $M$ answers that query.[2] Its goal differs fundamentally from capability circuit discovery: the former aims to trace and analyze the internal states of an LLM as it processes and responds to a user input (*local interpretations*), whereas the latter examines how an LLM implements particular algorithmic skills (*global interpretations*) (Bereska & Gavves, 2024).

### 3.2 EVALUATION OF QUERY CIRCUIT AND DISCOVERY METHOD

Similar to capability circuits, we aim to develop a faithfulness measure to quantify how well the discovered query circuit $C_q$ recovers the original LLM $M$'s performance on the query $q$, denoted as $F(\cdot) : C_q \to \mathbb{R}$. To evaluate a discovery method, we average $F(C_q)$ of different queries across a dataset $D$ (e.g., MMLU). Under an edge budget $N$, the performance of a discovery method is

$$\frac{1}{|D|} \sum_{q \in D} F(C_q), \tag{4}$$

where each $C_q$ has $N$ edges. A query circuit discovery method is more effective if, under varying $N$, it consistently produces query circuits with higher faithfulness scores than the counterpart. A straightforward choice of $F(\cdot)$ is inheriting the NFS metric, but we argue that it is unreliable and a suboptimal choice for evaluating query circuits and discovery methods, detailed in Section 3.3.1.

### 3.3 TECHNICAL CHALLENGES

#### 3.3.1 INSTABILITY OF NORMALIZED FAITHFULNESS SCORE ON GENERAL DATASETS

NFS has primarily been used to evaluate capability circuits on toy tasks with researcher-curated data (e.g., IOI). However, we find that it is not a reliable faithfulness measure on more general datasets of greater interest (e.g., MMLU). Figure 2a reports the NFS of three query circuits discovered by EAP-IG (Hanna et al., 2024) under varying edge budgets $N$. Unless otherwise stated, we adopt EAP-IG throughout this paper as the MIB benchmark (Mueller et al., 2025) finds it to be the most effective method. We randomly sample the three queries from the MMLU Marketing category. Llama-3.2-1B-Instruct (386713 edges) (Dubey et al., 2024) is the target model. The results show large fluctuations, with NFS values often exceeding 1 or dropping below 0 at different $N$. This instability undermines both the evaluation of circuit quality and the monitoring of discovery progress as $N$ increases (Miller et al., 2024). We therefore propose NDF as an alternative metric to evaluate the faithfulness of query circuits, detailed in Section 4.

#### 3.3.2 DEGRADATION OF CAPABILITY CIRCUIT DISCOVERY METHODS IN QUERY SETTINGS

We find that directly applying methods from capability circuit discovery to identify query circuits generally yields suboptimal results. Figure 2b presents a case study on the IOI dataset. In IOI, all queries require the same capability to generate correct tokens, allowing the construction of both a capability circuit for all queries and individual query circuits for comparison. With GPT-2 Small (32491 edges) (Radford et al., 2019) as the target LLM and an edge budget of $N = 1000$, the query circuit recovers on average less than 50% of GPT-2 Small's performance per query, while the capability circuit recovers roughly 65% of the model's overall performance on the dataset.

---

[2]The resulting query circuits can then be examined by human experts or external LLMs to interpret the roles of nodes and edges, though such interpretation lies beyond the scope of this paper.

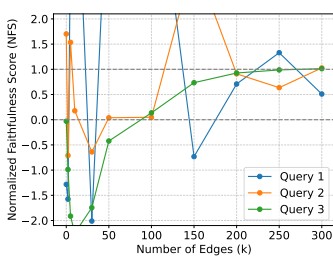 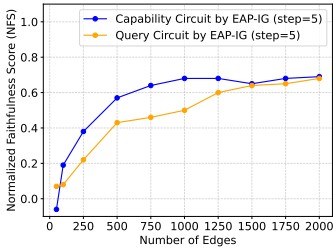 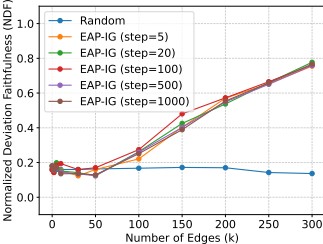

(a) Case study of three queries from MMLU Marketing showing **NFS' instability when applied to assess query circuits of different sizes.**

(b) Case study on IOI dataset showing **directly applying methods for capability circuits may yield less-faithful query circuits.**

(c) Case study on MMLU Astronomy showing **on complex queries, Many edges may be needed to recover non-trivial circuits.**

Figure 2: Technical challenges of query circuit discovery.

We attribute this degradation to two factors: (1) feature attribution suffers from *gradient noise* (Smilkov et al., 2017; Kapishnikov et al., 2021; Kim et al., 2019); and (2) the IE calculation (Equation 1) ignores combinatorial effects among edges (Shapley, 1953; Lundberg & Lee, 2017). For a given input, an edge transmitting irrelevant features may still exhibit non-zero gradients (and thus a non-zero attribution score) while contributing little when combined with others. This issue is less pronounced in capability circuit discovery, where $a_e$ is averaged over dataset $D$, diluting edges with only sporadically high scores.

### 3.3.3 HIGH EDGE BUDGET REQUIREMENTS FOR COMPLEX QUERIES

We find that directly applying capability-circuit methods to the query setting with complex queries requires far more edges to form a non-trivial circuit. Figure 2c presents a case study on MMLU Astronomy with Llama-3.2-1B-Instruct as the target model, using random edge selection as a baseline. On average, EAP-IG needs about $100k$ edges ($25.9\%$) to surpass the random baseline. This ineffectiveness may reflect either (i) the inherent need for many edges to capture natural-form MCQs or (ii) EAP-IG's inability to identify faithful circuits. Thus, we propose BoN sampling (Section 5) to test this hypothesis and attribute the issue to (ii) (Section 6). Moreover, Figure 2c shows that increasing the IG step does not improve discovery, consistent with our discussion (Section 3.3.2) on gradient noise and combinatorial effects, which cannot be resolved by refining single-edge IEs.

## 4 NORMALIZED DEVIATION FAITHFULNESS

### 4.1 DEFINITION AND PROPERTIES

The Normalized Deviation Faithfulness (NDF) of a query circuit $C_q$ is defined as

$$NDF(C_q) = 1 - \min\left(\left|\frac{L(M(q)) - L(C_q(q))}{L(M(q)) - L(M(q'))}\right|, 1\right), \tag{5}$$

which measures the performance deviation of a query circuit $C_q$ from the target LLM $M$, normalized by $M$'s performance gain from the corrupted query to the original query. NDF is derived from the integrated circuit-model distance (CMD) introduced by the MIB benchmark (Mueller et al., 2025), which quantifies the overall performance of circuit discovery methods. NDF differs from NFS in two key aspects. First, it is symmetric around $L(M(q))$, equally penalizing deviations above and below $M$'s performance on $q$. Second, NDF is bounded within the interval $[0, 1]$. $NDF(c_q) = 0$ if the performance deviation exceeds $M$'s performance gap between the original and corrupted query; $NDF(C_q) = 1$ when $C_q$ has the same performance as $M$. More discussions on the relations between NFS, NDF, and CMD are in Appendix C.

### 4.2 QUALITATIVE COMPARISON WITH NORMALIZED FAITHFULNESS SCORE

Table 1 presents three examples of query circuit faithfulness evaluated using NFS and NDF. These queries are MCQs from the MMLU Marketing dataset. Target LLM $M$ is Llama-3.2-1B-Instruct. Performance metric $L$ is the probability difference between the correct option and the average of

Table 1: **Examples of evaluating three query circuits from Figure 2a using NFS and NDF.** The corresponding queries are multiple-choice questions from the MMLU Marketing category.

| Query and Circuit Info | $L(M(q))$ | $L(M(q'))$ | $L(C(q))$ | NFS | NDF |
|---|---|---|---|---|---|
| Query 1 $\|C_q\| = 5k$ | -0.04 | -0.16 | 0.10 | 2.15 | 0.00 |
| Query 2 $\|C_q\| = 250k$ | 0.17 | 0.39 | 0.09 | 1.32 | 0.68 |
| Query 3 $\|C_q\| = 5k$ | 0.96 | 0.53 | -0.13 | -1.57 | 0.00 |

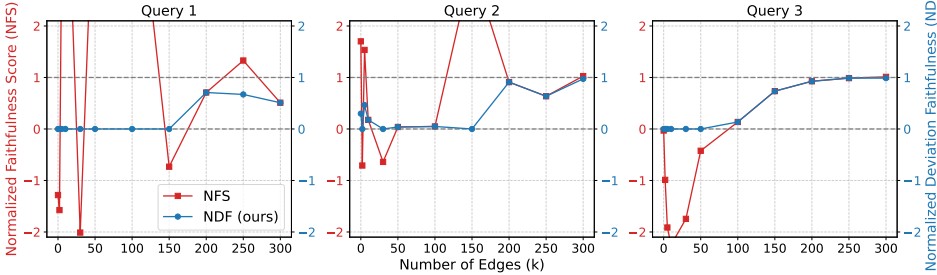

Figure 3: Complete evaluation results for the three queries adopted in Figure 2a and Table 1.

the three incorrect options. NFS exhibits numerical instability in several scenarios—for example, when $M$'s performance gap between $q$ and $q'$ is small (as in Query 1), or when $M$ achieves non-zero performance on $q'$ (e.g., due to position bias (Zheng et al., 2024), as in Query 3). In contrast, our proposed NDF, which measures the faithfulness of $C_q$ as its normalized performance deviation from $M$, provides a more stable and reliable evaluation. Accordingly, we adopt NDF as the primary metric for all subsequent experiments. Figure 3 presents complete evaluation results for three queries, further supporting this choice, with additional results provided in Appendix B.1.

## 5 BEST-OF-N SAMPLING FOR QUERY CIRCUIT DISCOVERY

In this section, we introduce Best-of-N (BoN) sampling for query circuit discovery. We first present our motivation—a preliminary observation of circuit discovery on a query and its paraphrases in Section 5.1, introduce BoN in Section 5.2, and then detail two extensions: (1) interpolated BoN (iBoN) in Section 5.3 and (2) BoN with Constraint-adaptive Score Matrix (BoN-CSM) in Section 5.4.

### 5.1 OBSERVATION: FAILURE IN A QUERY, SUCCESS IN ITS PARAPHRASES

We find that *while the circuit discovered on the original query may fail to faithfully recover model performance, circuits discovered on its paraphrases can succeed.* Figure 4 illustrates this with a query $q$ from the IOI dataset using GPT-2 Small as the target model. Although EAP-IG consistently fails to identify a faithful circuit for $q$ directly, applying it to randomly selected paraphrases of $q$ can find small, faithful ones.

We argue that, due to gradient noise and the neglect of combinatorial effects (Section 3.3.2), edge scoring based on Equations 1 and 2 for a query $q$ can only capture coarse score patterns—represented as a score matrix $S$ (examples are in Appendix B.2)—that roughly separate crucial from trivial edges, but are not precise enough to consistently select a set of edges that forms a faithful circuit. Score matrices from paraphrases can be viewed as perturbations of $S$: while

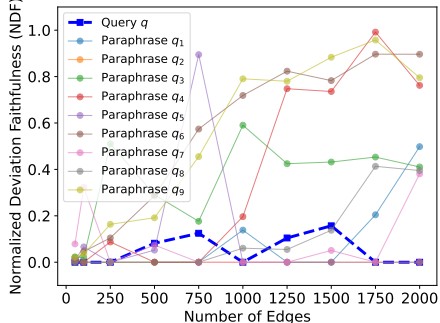

Figure 4: A case study on IOI dataset. **Circuits discovered by the original input query's paraphrases may recover model performance on the query.**

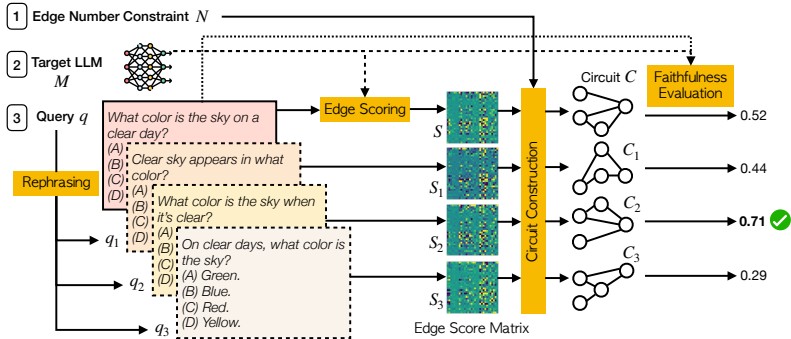

Figure 5: **The pipeline of Best-of-N sampling for discovering a faithful query circuit of $N$ edges for an input query $q$,** for which it generates $p$ paraphrases. $p = 3$ in this illustration.

---

**Algorithm 1** iBoN

**Input:** Circuits (edge sets) $\{E_1, \dots, E_k\}$ with size in ascending order and edge number constraint $N$.

**Output:** An edge set (circuit) $E$

1: Initialize $E$ as an empty edge set.
2: Find the largest $i$ s.t. $|E_i| < N$.
3: $E \leftarrow E_i$.
4: $K := N - |E|$.
5: $E_{i+1}^{1:K} :=$ top-$K$ edges of $E_{i+1}$ not in $E_i$.
6: $E \leftarrow E \cup E_{i+1}^{1:K}$.
7: **return** $E$

---

**Algorithm 2** BoN-CSM

**Input:** Circuits (edge sets) $\{E_1, \dots, E_k\}$ with size in ascending order.

**Output:** Score matrix $S$ and tier matrix $T$

1: Initialize $S$, $T$, and boolean matrix $B$.
2: **for** $i, E_i$ in enumerate($\{E_1, \dots, E_k\}$) **do**
3:     **for** $e$ in $E_i$ **do**
4:         $a_e :=$ attribution score of $e$.
5:         $(j, k) :=$ score matrix index of $e$.
6:         **if** $B(j, k)$ is not *True* **then**
7:             $S(j, k) \leftarrow a_e; T(j, k) \leftarrow i$
8:             $B(j, k) \leftarrow$ *True*.
9:         **end if**
10:     **end for**
11: **end for**
12: **return** $S$ and $T$

---

they share similar patterns, small differences in edge scores can considerably alter which edges are selected. In this case, finding a faithful circuit within the model is akin to a lottery (Frankle & Carbin, 2019): circuits discovered by the original query and its paraphrases are "tickets," and the one that successfully recovers the model performance on the query is the "winning ticket."

## 5.2 BEST-OF-N SAMPLING

Based on the observation in Section 5.1, we introduce Best-of-N (BoN) sampling for query circuit discovery. As shown in Figure 5, to find a "winning ticket", BoN first generates $p$ paraphrases of the original query $q$, denoted as $\{q_1, ..., q_p\}$ (e.g., p = 3 in Figure 5). Then, it calculates edge importance scores $a_e$ by each of $\{q, q_1, ..., q_p\}$, represented as edge score matrices $\{S, S_1, ..., S_p\}$ (see Appendix B.2 for examples of these matrices). Finally, it leverages $\{S, S_1, ..., S_p\}$ to form $p + 1$ circuits, measure their faithfulness score, and select the one with the highest score.

Steps 1 and 2 are required only once when constructing circuits with different edge budgets $N$. However, step 3 needs $p + 1$ forward passes of the target LLM $M$ to identify the best circuit for a given $N$, which becomes a time bottleneck if one aims to construct circuits of many sizes. To address this issue, we introduce two simple extensions of BoN: iBoN and BoN-CSM. Both build on BoN-discovered faithful circuits to accelerate the discovery of circuits of varying sizes.

## 5.3 INTERPOLATED BEST-OF-N

Algorithm 1 shows the procedure of interpolated Best-of-N (iBoN), with circuits denoted as their edge sets for simplicity. iBoN *interpolates* between two previously discovered faithful circuits to efficiently form a new one without an LLM. Assume one has applied BoN to discover $k$ circuits $\{E_1, ..., E_k\}$ with different edge counts $N$ (WLOG assume $|E_i| < |E_j|$ if $i < j$). Then, for a new $N$ of interest where $N \notin \{|E_1|, ..., |E_k|\}$ and $|E_1| < N < |E_k|$, iBoN constructs an intermediate

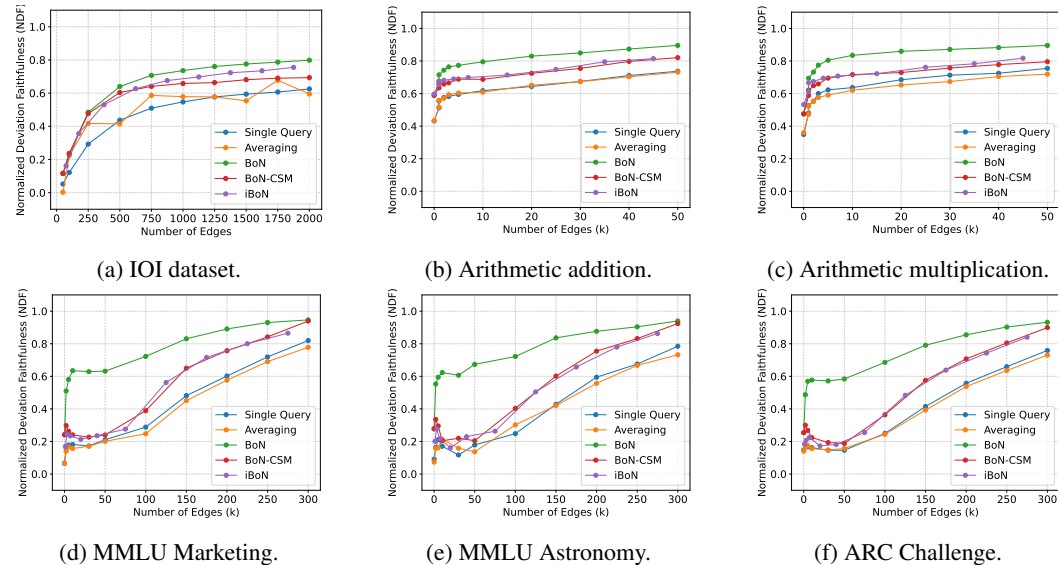

(a) IOI dataset.    (b) Arithmetic addition.    (c) Arithmetic multiplication.

(d) MMLU Marketing.    (e) MMLU Astronomy.    (f) ARC Challenge.

Figure 6: **Main results of BoN sampling for query circuit discovery.** BoN substantially outperforms all other methods. Although iBoN and BoN-CSM, two fast approximations to BoN, perform worse than it, they still clearly exceed both baselines.

circuit by augmenting the best available smaller circuit with additional high-scoring edges from a larger one that are not already included.

## 5.4 BON WITH CONSTRAINT-ADAPTIVE SCORE MATRIX

BoN with Constraint-adaptive Score Matrix (BoN-CSM, Algorithm 2) leverages all $k$ previously discovered circuits $\{E_1, ..., E_k\}$ of different edge budgets (i.e., constraints) to establish a score matrix $S$ and a tier matrix $T$, which are then used to efficiently form new circuits. It first initializes $S$, $T$, and an auxiliary index matrix $B$. Starting from the smallest circuit $E_1$, it iteratively records each edge $e$'s score $a_e$ to $S$ and the current circuit index (e.g., $i$ for $E_i$) to $T$, while using $B$ to avoid duplicate entries. In this way, $S$ and $T$ determine the importance scores and priorities of all edges that have been identified in $\{E_1, \ldots, E_k\}$. When constructing a new circuit of size $N$ where $|E_1| < N < |E_k|$, it first sorts all edges in $S$ by their tiers in $T$ to prioritize those from smaller (i.e., high-tier) circuits. It further sorts the edges within each tier by their importance scores in $S$. Then, it selects top-$N$ edges from this tier-then-score order to form the circuit, requiring no additional LLM forward pass.

## 6 EXPERIMENTS

### 6.1 EXPERIMENTAL SETUP

We conduct query circuit discovery with BoN sampling on IOI (Wang et al., 2023), arithmetic addition, arithmetic multiplication, ARC Challenge (Clark et al., 2018), and nine categories of MMLU (Hendrycks et al., 2021). Performances of circuits are averaged over all queries in the datasets. For each IOI query, we randomly select nine other queries from the dataset as its paraphrases (i.e., $p = 9$). For arithmetic addition and multiplication, paraphrases are produced by permuting the operands. For ARC Challenge and MMLU, we use GPT-4o (Hurst et al., 2024) to generate nine paraphrases of the question stem. We adopt EAP-IG (step $m = 20$) as the backbone method to estimate edge scores and use greedy selection to construct edges. We consider two baselines: (i) estimate each edge's importance score $a_e$ simply on that query; and (ii) estimate $a_e$ as the average over the query and its paraphrases. Unless otherwise specified, we adopt GPT-2 Small (32491 edges) as the target LLM for IOI and Llama-3.2-1B-Instruct (386713 edges) for all other tasks. Refer to Appendix A for detailed experimental setup and design choices.

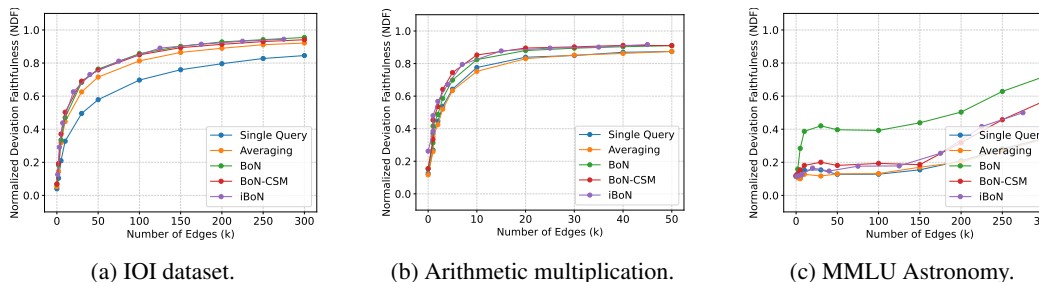

(a) IOI dataset.

(b) Arithmetic multiplication.

(c) MMLU Astronomy.

Figure 7: **Scaling BoN sampling for query circuit discovery to larger models (GPT-2 XL for IOI; Llama-3-8B-Instruct for arithmetic multiplication and MMLU astronomy).** BoN, iBoN, and BoN-CSM still consistently outperform both baselines.

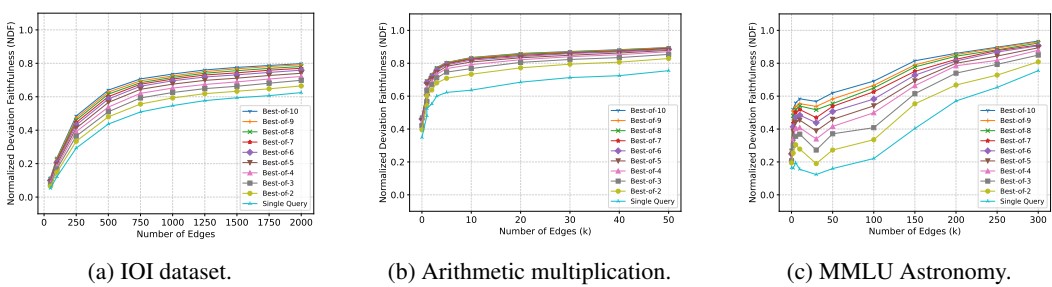

(a) IOI dataset.

(b) Arithmetic multiplication.

(c) MMLU Astronomy.

Figure 8: **Performance of BoN sampling with different numbers of paraphrases**. As BoN selects the most faithful circuit, its performance exhibits monotonically increasing yet diminishing returns.

## 6.2 MAIN RESULTS

Figure 6 presents the results of query circuit discovery across different tasks (complete MMLU results are in Appendix B.3). BoN, iBoN, and BoN-CSM consistently outperform both baseline (i) (*Single Query*) and baseline (ii) (*Averaging*). In particular, BoN surpasses other methods by a large margin, requiring orders of magnitude fewer edges to construct non-trivial query circuits. In MMLU, a circuit with only $5k$ edges (1.3% of Llama-3.2-1B-Instruct's all edges) achieves an average NDF of 0.6, whereas vanilla EAP-IG (single-query baseline) suggests that $200k$ (51.7%) edges are needed to reach that level. The results advance recent findings of input-dependent activation sparsity (Li et al., 2023; Szatkowski et al., 2025) to circuitry sparsity, demonstrating the promise of finding compact, critical information flow within an LLM responsible for answering an input query. Notably, the averaging method does not perform better than the single-query one. This is probably because, while prioritizing edges scored high on both the input query and its paraphrases can be beneficial, it simultaneously downweights edges that matter only to the original query, which may be crucial for constructing a faithful circuit for that specific query.

We further scale the target models to GPT-2 XL (1.5B; 2235025 edges) for IOI and Llama-3-8B-Instruct (1592881 edges) for arithmetic multiplication and MMLU astronomy, as shown in Figure 7. Our methods continue to consistently outperform both baselines. Notably, on average, BoN discovers a 5,000-edge query circuit (0.3% of all edges) in Llama-3-8B-Instruct that reconstructs 0.4 NDF for the input query. In contrast, vanilla EAP-IG misleadingly suggests that even a 300k-edge circuit (18% of the model) cannot achieve this level of faithfulness, giving a false impression that the underlying computational mechanism is much denser than it actually is. Appendix B provides many more additional experiments, such as other variants of BoN sampling (Appendix B.7).

## 6.3 ABLATION STUDIES AND RUNTIME ANALYSIS

Figure 8 shows the performance of BoN with different numbers of paraphrases. Since BoN selects the most faithful circuit, its performance increases monotonically as the number of paraphrases grows. However, the gains diminish because additional paraphrases often provide overlapping or redundant information, making it less likely for them to contribute new high-quality circuits.

Table 2: Average runtime of EAP-IG and BoN for discovering and evaluating a query circuit.

| Method | EAP-IG | | | | | BoN | | |
|---|---|---|---|---|---|---|---|---|
| Parameter | 5 | 20 | 100 | 500 | 1000 | 1 | 4 | 9 |
| Per-query Runtime $(s)$ | 4.3 | 9.5 | 27.9 | 120.2 | 237.5 | 25.4 | 66.0 | 132.0 |

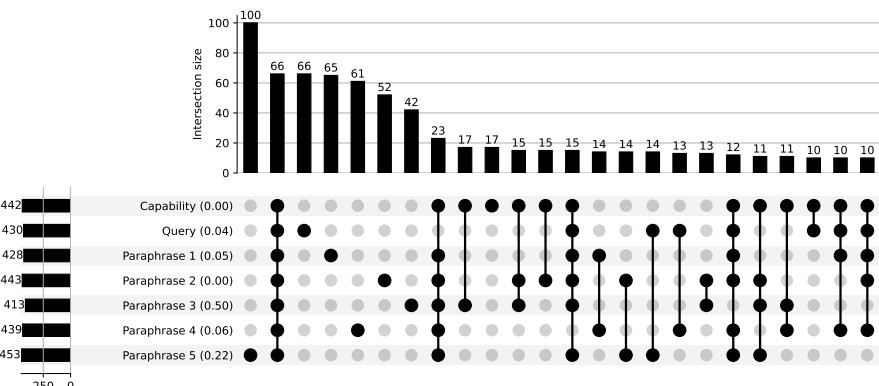

Figure 9: **UpSet plot of the capability circuit and query circuits of a randomly selected query discovered by it or its paraphrases**. A total of 66 edges are shared across all circuits.

Table 2 reports the average runtime for discovering and evaluating query circuits on MMLU Astronomy. For each query, we identify and evaluate 11 circuits of varying sizes as in Figure 6. The reported runtime is averaged over the 11 circuits and then over all 152 samples. We compare EAP-IG with varying IG steps and BoN with different numbers of additional paraphrases. Runtime of BoN with nine paraphrases is slightly longer than that of EAP-IG with 500 steps, while EAP-IG consistently yields suboptimal performance in query circuit discovery even with 1000 steps (Figure 2c).

### 6.4 CIRCUIT VARIANCES AND EXISTENCE OF SHARED SUB-CIRCUIT

Using the IOI dataset, we further examine the relationship between query circuits and the capability circuit (IEs averaged over 1000 queries). Specifically, we investigate whether BoN sampling (1) merely produces unrelated disjoint circuits where some coincidentally output the correct token, or (2) discovers variants of a common mechanism that preserve a shared set of critical edges (i.e., shared sub-circuit), regardless of how the query is phrased.

Figure 9 provides preliminary support for (2). It shows the UpSet plot of the capability circuit and the query circuits of a randomly selected query (edge budget $N = 500$). Each row corresponds to a circuit, with its number of edges and NDF score; each column indicates the number of edges shared among the circuits in black dots. Notably, 66 edges appear in all circuits, supporting the existence of a shared sub-circuit. We also observe 23 edges (8th column) missing only from the circuit derived from the original query—meaning that relying solely on the original query would fail to recover these edges. Additional qualitative and quantitative evidence for (2) is provided in Appendix B.9.

## 7 CONCLUSION

We introduce query circuit discovery, the task of identifying the information flow within an LLM responsible for answering an input query. We first formalize the task and distinguish it from capability circuit discovery, then discuss its technical challenges and propose methods to address them. Specifically, we introduce NDF as a more reliable evaluation metric for assessing query circuits and explore BoN sampling for discovering faithful query circuits. Experiments show that even for complex queries, compact sub-networks within the LLM can still recover much of the model's behavior, establishing BoN as a useful technique and query circuit discovery as a promising direction for explaining LLM decisions. We discuss limitations and future work in Appendix D.

## 8 ETHICS STATEMENT

This work adheres to the ICLR Code of Ethics and contains no confidential data, sensitive content, or experiments involving human subjects. We note that mechanistic interpretability methods, including ours, should be used with caution to avoid incorrect interpretations that could lead to adverse consequences.

## 9 REPRODUCIBILITY STATEMENT

We confirm that this work follows a controlled experimental process to ensure the reproducibility of all figures and tables in the main text. All experiment scripts are executed with a fixed random seed (2025) across all packages. For MCQ paraphrase generation, GPT-4o's temperature is fixed at 0 to guarantee reproducibility, and all generated paraphrases are recorded and will be released. For arithmetic tasks, paraphrase generation involves randomness, but this is controlled by the same fixed seed (2025). The circuit discovery process itself is deterministic and does not involve randomness. The code will be publicly released upon acceptance.

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

# SUPPLEMENTARY MATERIAL

## A  DETAILED EXPERIMENTAL SETUP

This section serves as an extension of Section 6 to provide a more detailed experimental setup, design choices, and their implications.

**Datasets.**  We conduct query circuit discovery on the IOI dataset, arithmetic addition, arithmetic multiplication, ARC Challenge, and nine categories from MMLU. We randomly select the nine categories from all 52 categories in which Claude 3.5 Sonnet (2024/10/22 version) (Antropic, 2025) achieves at least 95% accuracy on the Stanford HELM MMLU leaderboard (Liang et al., 2023). The IOI dataset follows Hanna et al. (2024)'s implementation and has 1000 queries. The arithmetic addition and multiplication datasets each consist of 500 queries, covering operands of length 2–5 (125 queries per length). Each query's answer is an integer less than 1000. These datasets are more challenging than the two-operand arithmetic addition and subtraction tasks used in the MIB benchmark. For ARC Challenge, we adopt the test split (1172 MCQs). The nine selected MMLU categories are: marketing (234 MCQs), professional medicine (272 MCQs), astronomy (152 MCQs), college biology (144 MCQs), high school computer science (100 MCQs), logical fallacies (163 MCQs), nutrition (306 MCQs), international law (121 MCQs), and management (103 MCQs).

**Paraphrases.**  For each IOI query, we randomly select nine other queries from the dataset as paraphrases since every query in the IOI dataset is itself a word-swapped variant of another. In arithmetic addition and multiplication, paraphrases are generated by permuting the operands. The number of available paraphrases varies with the number of operands, but we limit each query to at most nine paraphrases. For ARC Challenge and MMLU, we use GPT-4o to generate nine paraphrases of the question stem.

**Corrupted queries.**  For IOI, a corrupted query is constructed by replacing the repeated name in the original query with a third name, as described in Section 2.2.1. For arithmetic addition and multiplication, the corrupted query is another sample with the same number of operands but a different answer. For ARC Challenge and MMLU, the corrupted query is created by replacing the question stem with *"Which is the most possible answer?"*. Table A3 shows the examples of original and corrupted queries.

Note that the form of corrupted queries directly affects both the functionality and interpretation of the discovered circuits. For MCQs, under our proposed corruption strategy, the discovered edges capture critical interactions between the stem and the choices. This arises because such interactions are present in the original query but absent in the corrupted one. By contrast, the MIB benchmark, as an early attempt at circuit discovery for MCQs, constructs corrupted queries through semantics-irrelevant rephrasing—for example, changing option IDs from (A), (B), (C), (D) to (1), (2), (3), (4). Under this formulation, the discovered circuits primarily contain edges associated with ID matching rather than meaningful stem–choice reasoning and factual retrieval.

**Baselines.**  We adopt EAP-IG as the backbone method to score edges since it is one of the most effective current approaches. The original EAP-IG implementation employs a Dijkstra-like iterative construction introduced in Section 2.2.1. Our replications show that it achieves comparable performance to greedy selection but with higher runtime (see Appendix B.5). As a result, we adopt greedy selection to construct the circuit after edge scoring. The IG step is set to 20 throughout the experiments. Two baselines are: (i) applying EAP-IG directly to each original query, i.e., estimating each edge's IE on that query; and (ii) applying EAP-IG to estimate each edge's IE averaged over the original query and its paraphrases. The latter is exactly the way methods in capability circuit discovery score edges for capability circuit construction.

**Target Model and Performance Metric.**  For IOI, we use GPT-2 Small (32491 edges) as the target LLM, following prior work; for all other tasks, we use Llama-3.2-1B-Instruct (386713 edges). For performance metric $L$, we adopt logit difference as it provides a more natural unit for transformers than probability difference (Heimersheim & Nanda, 2024). Specifically, for IOI, the logit difference between the correct and incorrect name is adopted. For arithmetic addition and multiplication, the logit difference between the correct and corrupted answers is used. For ARC Challenge and MMLU, we consider the logit difference between the correct option and the average of the incorrect ones. Performances of circuits are averaged over all queries in the datasets.

Table A3: Examples of original and corrupted queries from the datasets used in this paper. For arithmetic questions, the corrupted query has the same number of operands but a different answer. For MCQs, the corrupted query preserves the options, but the question stem is replaced with a prompt that simply asks the model to choose one.

| Dataset | Clean Query | Corrupted Query |
|---|---|---|
| IOI | When Amy and Laura got a snack at the house, Laura decided to give it to | When Amy and Laura got a snack at the house, Nicholas decided to give it to |
| Arithmetic Add. | 41+260+303+48+87= | 11+52+23+18+6= |
| Arithmetic Mul. | 7*2*2*3*10= | 2*2*14*4*4= |
| MMLU | What is true for a type-Ia (""type one-a"") supernova?
(A) This type occurs in binary systems.
(B) This type occurs in young galaxies.
(C) This type produces gamma-ray bursts.
(D) This type produces high amounts of X-rays.
Answer: ( | Which is the most possible answer?
(A) This type occurs in binary systems.
(B) This type occurs in young galaxies.
(C) This type produces gamma-ray bursts.
(D) This type produces high amounts of X-rays.
Answer: ( |
| ARC Challenge | Two girls are pulling on opposite ends of a thick rope. Both girls pull on the rope with the same force but in opposite directions. If both girls continue to pull with the same force, what will most likely happen?
(A) One girl will pull the other toward her.
(B) Both girls will stay in the same place.
(C) Gravity will cause the rope to sag.
(D) The rope will break.
Answer: ( | Which is the most possible answer?
(A) One girl will pull the other toward her.
(B) Both girls will stay in the same place.
(C) Gravity will cause the rope to sag.
(D) The rope will break.
Answer: ( |

**Edge budgets.** When testing all methods except iBoN: For IOI, we set $N \in \{50, 100, 250, 500, 750, 1k, 1.25k, 1.5k, 1.75k, 2k\}$; For arithmetic addition and arithmetic multiplication, we use $N \in \{500, 1k, 1.5k, 2k, 3k, 5k, 10k, 20k, 30k, 40k, 50k\}$; For ARC Challenge and MMLU, we consider $N \in \{500, 2k, 5k, 10k, 30k, 50k, 100k, 150k, 200k, 250k, 300k\}$. When testing iBoN, we adopt interpolated budgets because iBoN will produce the same performance as BoN if it has the same edge budget. Specifically, for iBoN on IOI, we set $N \in \{75, 175, 375, 625, 875, 1.125k, 1.375k, 1.625k, 1.875k\}$; For arithmetic addition and arithmetic multiplication, we use $N \in \{750, 1.25k, 1.75k, 2.5k, 4k, 7.5k, 15k, 25k, 35k, 45k\}$; For ARC Challenge and MMLU, we consider $N \in \{1.25k, 3.5k, 7.5k, 20k, 40k, 75k, 125k, 175k, 225k, 275k\}$.

## B  MORE EXPERIMENTAL RESULTS

### B.1  MORE COMPARISONS OF QUERY CIRCUIT EVALUATION RESULTS BY NFS AND NDF

Figure A10 presents 18 examples of query circuit evaluation using NDF and NFS. The queries are the first 18 samples from MMLU Marketing. These results show that our proposed NDF provides a more stable assessment and can track discovery progress as the circuits become larger.

### B.2  EXAMPLES OF SCORE MATRIX

Figure A11 and Figure A12 show the score matrices for the first query in the IOI dataset and MMLU Astronomy, along with their nine paraphrases. The target LLM is Llama-3.2-1B-Instruct. For clear visualization, we visualize only a subset of the score matrix—specifically, edges within layers 7–10 (out of the 16 layers). The discovery method is EAP-IG with step size $m = 20$. The matrices are not square because, in practice, when parent nodes are attention heads, they will be split into query, key, and value heads by computing the gradients flowing through each.

The score matrices of the original query and its paraphrases exhibit similar patterns: the scores of certain edges remain high, while others are consistently low. On the other hand, score patterns between two different query types (IOI vs. MMLU MCQs) are more distinct. As argued and experimented in the main paper, the score pattern of a single query, though meaningful, is often not sufficiently precise for constructing faithful query circuits—motivating our exploration of paraphrase-based discovery to generate slightly different yet pattern-aligned score matrices.

### B.3  COMPLETE RESULTS OF NINE MMLU CATEGORIES

Figure A13 presents the full results for nine randomly selected MMLU categories. BoN consistently achieves around 0.6 NFS using only 5000 of the 386713 edges (1.3%) in Llama-3.2-1B-Instruct. While iBoN and BoN-CSM do not yield circuits that are as faithful yet sparse as BoN, they still outperform the baseline methods clearly and consistently.

### B.4  FAITHFULNESS SCORES OF COMPLEMENT CIRCUITS

Figure A14 shows the NDF scores of complement circuits $C^c := E \setminus E_q$, where $E$ is the LLM's edge set and $E_q$ the query circuit's. Low, near-random faithfulness scores of complement circuits indicate that critical information flow indeed resides within the query circuits.

This experiment reflects a standard practice in circuit-discovery studies: following prior work (e.g., Figure 3 in feature circuits (Marks et al., 2025)), we adopt counterfactual evaluations by measuring the faithfulness of both a circuit $C$ (Figure 6) and its complement $C^c$ (Figure A14). Faithfulness of $C$ corresponds to a sufficiency test—whether $C$ alone can reconstruct model behavior; whereas faithfulness of the complement $C^c$ acts as a necessity test: if $C$ truly contains the information required for the model to parse the input and generate the correct response, then ablating $C$ should break model performance, yielding low faithfulness for $C^c$. Consistent with Marks et al. (2025), we observe uniformly low NDFs across methods for complement circuits, supporting the necessity of the discovered query circuits.

This analysis also clarifies why Figure 6 shows performance differences across discovery methods, while complement circuits in Figure A14 remain uniformly unfaithful: low NDFs for both a query circuit and its complement simply indicate that neither alone forms a precise, self-sufficient combination of edges capable of reconstructing the full model behavior—only the true underlying circuit does.

Finally, for MMLU and ARC Challenge, complement circuits (Figure A14) and randomly constructed circuits (Figure 2c) both yield NDF scores around 0.1–0.2, rather than 0. This is because we compute logit differences only among the options, and both original and corrupted queries still contain signals that lead the model to distribute logits across option IDs. As a result, even if a circuit fails to capture model performance well, its performance deviations from the original LLM may still be smaller than the gap between the original and corrupted queries in a few samples.

Table A4: Runtime of greedy selection and Dijkstra-like iterative construction for forming circuits based on edge scores. The latter's runtime increases with respect to the number of edges $N$.

| Construction Method | Greedy Selection | | | Dijkstra-like Iteration | | |
|---|---|---|---|---|---|---|
| edge number $N$ | $10k$ | $100k$ | $300k$ | $10k$ | $100k$ | $300k$ |
| Per-circuit Runtime ($s$) | $< 0.1$ | $< 0.1$ | $< 0.1$ | 23.9 | 274.8 | 729.9 |

## B.5 COMPARISON OF GREEDY SELECTION AND DIJKSTRA-LIKE CONSTRUCTION

We compare the efficiency and effectiveness of greedy selection and Dijkstra-like construction for circuit discovery. Figure A15 presents results on the IOI, GT, and Gender-bias datasets with GPT-2 Small as the target model. The two methods achieve similar performance across all three datasets. Table A4 further reports the runtime of the two methods for constructing a circuit in Llama-3.2-1B-Instruct after obtaining the score matrix $S$. The Dijkstra-like iterative construction incurs substantially higher runtime as the edge budget $N$ increases. In contrast, greedy selection requires constant time regardless of circuit size, so we adopt it throughout this work.

## B.6 RUNTIME OF ACTIVATION AND ATTRIBUTION PATCHING IN QUERY SETTING

We present additional runtime analysis of activation patching and attribution patching methods, two major categories of circuit discovery within the model, in the setting of query circuits. For the former, we use ACDC; for the latter, we use EAP and EAP-IG. As shown in Figure A16, ACDC takes about 18000 seconds (5 hours) on an NVIDIA A100 to discover a circuit for a query in GPT-2 Large on the IOI dataset. This is because edge activation patching requires two LLM forward passes per edge, as discussed in Section 2.2.1. In contrast, EAP (Syed et al., 2024) and EAP-IG (Hanna et al., 2024), two representative attribution patching methods, take less than 10 seconds as they score all edges at once.

Since LLM systems process numerous queries per day (Chatterji et al., 2025), it is important that the query circuit discovery methods to trace and explain model decisions are scalable. As a result, we do not consider ACDC as a backbone method for query circuit discovery in this paper.

## B.7 ADDITIONAL VARIANTS OF BoN SAMPLING

We introduce and investigate three additional variants of BoN sampling for query circuit discovery here. (i) BoN-GP (Gaussian Perturbation): add Gaussian noise $G(0, \sigma^2)$ to the score matrix $S$ from the original query to alter edge selection. Repeating this $p$ times yields $S, S_1, \ldots, S_p$, and BoN sampling is performed over the 10 resulting circuits under edge budget $N$. (ii) BoN-ER (Edge Replacement): given a circuit at budget $N$, randomly replace $t \times 100\%$ of its edges with unused ones for $p$ trials, then select the best circuit among these and the original. (iii) BoN-Random: randomly sample $N$ edges to form a circuit, repeat 10 times, and take the best. Our main method, BoN-Para., instead uses $p$ paraphrases to produce additional $p$ different score matrices.

Figure A17 presents results on IOI and MMLU Astronomy, with $p = 9$ for all methods. For BoN-GP, we set $\sigma \in \{0.01, 0.001\}$; for BoN-ER, $t \in \{0.1, 0.3\}$. Both BoN-GP and BoN-ER show potential to discover small, faithful query circuits in MMLU Astronomy, but our main method, BoN-Para. (semantics-preserving score matrix perturbation), remains superior. Notably, BoN-Random remains stuck near 0.2 NDF—similar to the single-query baseline—until circuit size exceeds $50k$ edges, after which it begins to outperform the baseline up to about $200k$ edges. This likely occurs because, as more edges are added, random selection has a higher chance of including all critical edges and forming a large circuit that recovers model performance, a chance that is further amplified by BoN sampling.

## B.8 QUERY CIRCUIT DISCOVERY EVALUATED BY NFS

Figure A18 reports query circuit discovery results on the complete IOI and MMLU Astronomy datasets, evaluated using NFS instead of our proposed NDF metric. On IOI, the researcher-curated

toy dataset, NFS scores mostly remain within $[0, 1]$. In contrast, for MMLU Astronomy, NFS scores fluctuate widely even after averaging over all 152 samples, making it difficult to track discovery progress and undermining confidence in circuit faithfulness as measured by NFS. This motivates our proposal of NDF as a more robust and reliable alternative, as shown and discussed in the main paper.

### B.9 MORE ANALYSIS ON CAPABILITY CIRCUIT AND QUERY CIRCUIT

Here, we provide additional experiments analyzing the relationship between the capability circuit and query circuits. In Figures A19a and A19b, the averaged Jaccard similarity of around 0.3 indicates non-trivial edge overlap. Figure A19c further shows that 30–50% of edges in query circuits also appear in the capability circuit. In Figure A20, we show UpSet plots—analogous to Figure 9—for five additional queries, all exhibiting substantial shared edges. All queries are randomly selected with seed 2025, as stated in Section 9. Finally, Figure A5 visualizes the full set of seven circuits analyzed in Figure 9 (the capability circuit and the circuits derived from the original query and five paraphrases). Nodes and edges shared by all circuits are marked in red; others in green. These shared components constitute a common sub-circuit present across all circuit variants for the IOI task, regardless of query phrasing or whether IEs are averaged over many IOI queries.

Table A5: Complete plots of the seven circuits analyzed in Figure 9. Shared nodes and edges are shown in red; others in green. These shared components constitute the sub-circuit common to all seven circuits.

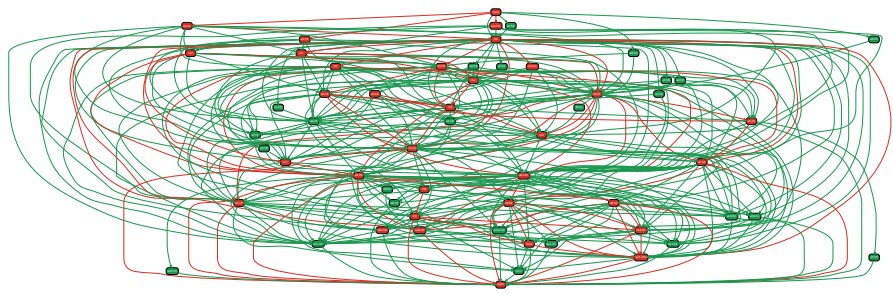

(a) Capability circuit.

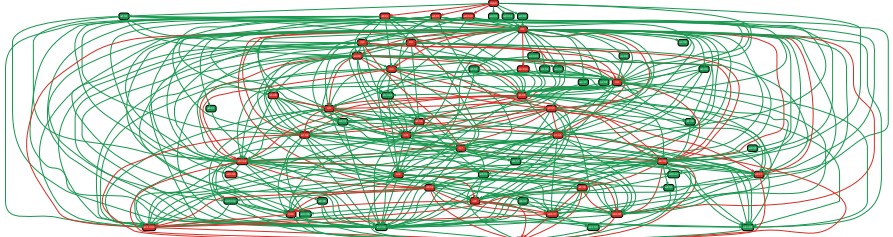

(b) Query circuit by the original query.

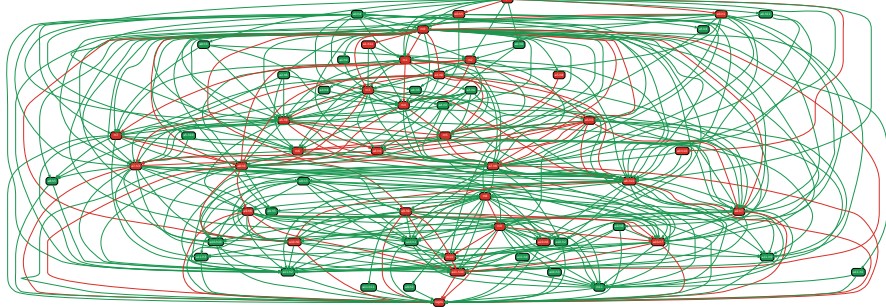

(c) Query circuit by paraphrase 1.

Table A5: continued

(d) Query circuit by paraphrase 2.

(e) Query circuit by paraphrase 3.

(f) Query circuit by paraphrase 4.

(g) Query circuit by paraphrase 5.

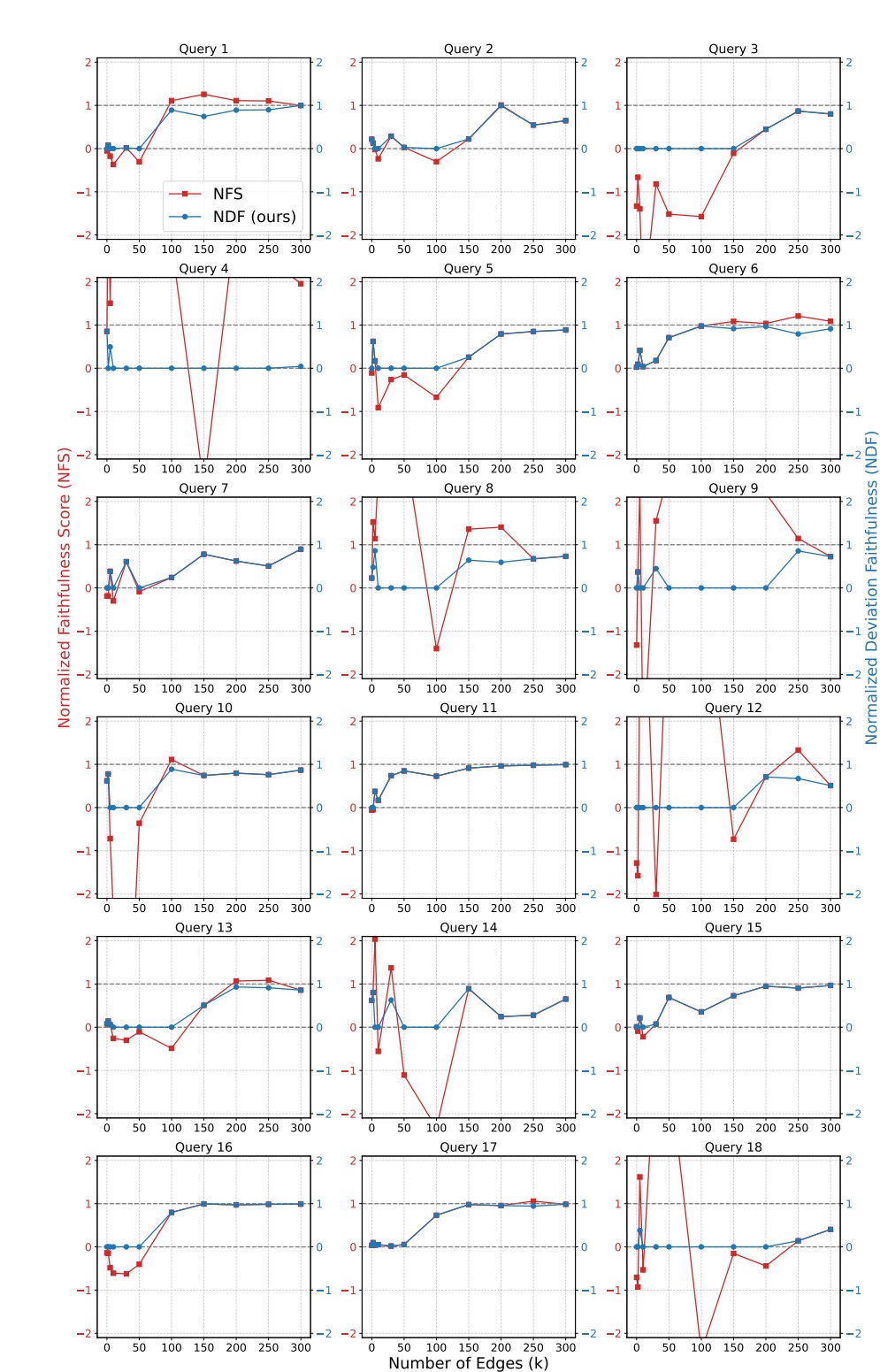

Figure A10: More query circuit evaluation results using NFS and our proposed NDF, which provides more stable evaluation and can better track the discovery progress as the circuit size grows.

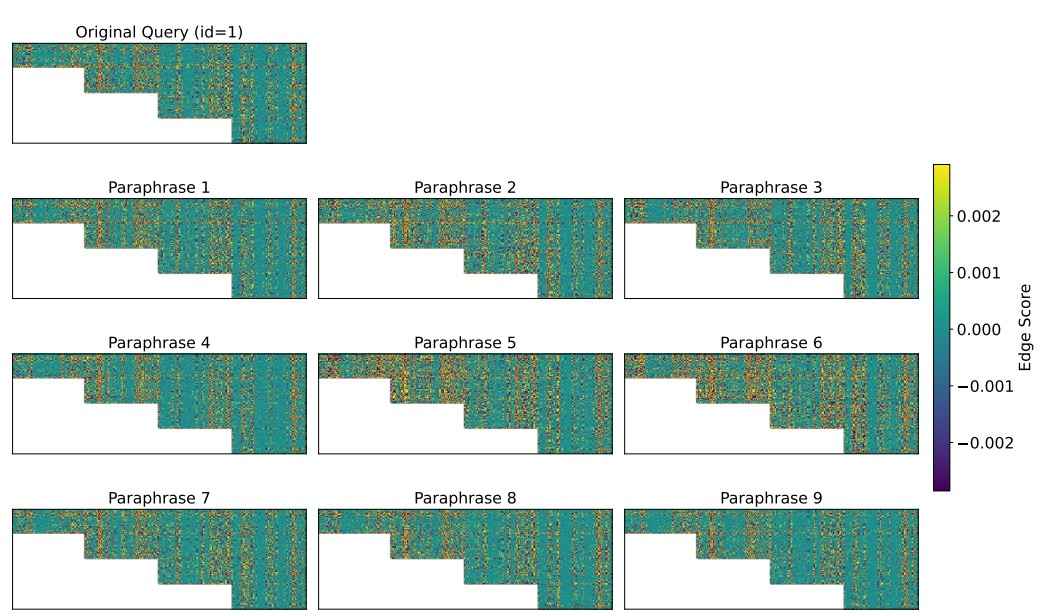

Figure A11: Edge score matrices of a query and its nine randomly selected paraphrases in IOI. EAP-IG is used to calculate the score of each edge (i.e., each entry in the matrix). These matrices share similar patterns.

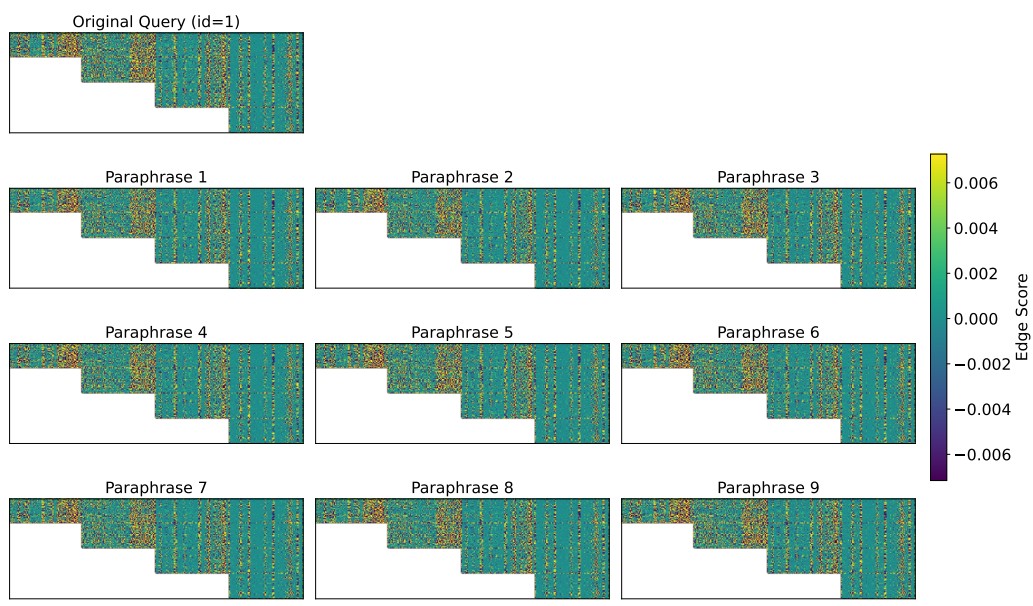

Figure A12: Edge score matrices of a query and its nine paraphrases in MMLU Astronomy. EAP-IG is used to calculate the score of each edge (i.e., each entry in the matrix). These matrices share similar patterns, and are dissimilar to those in Figure A11.

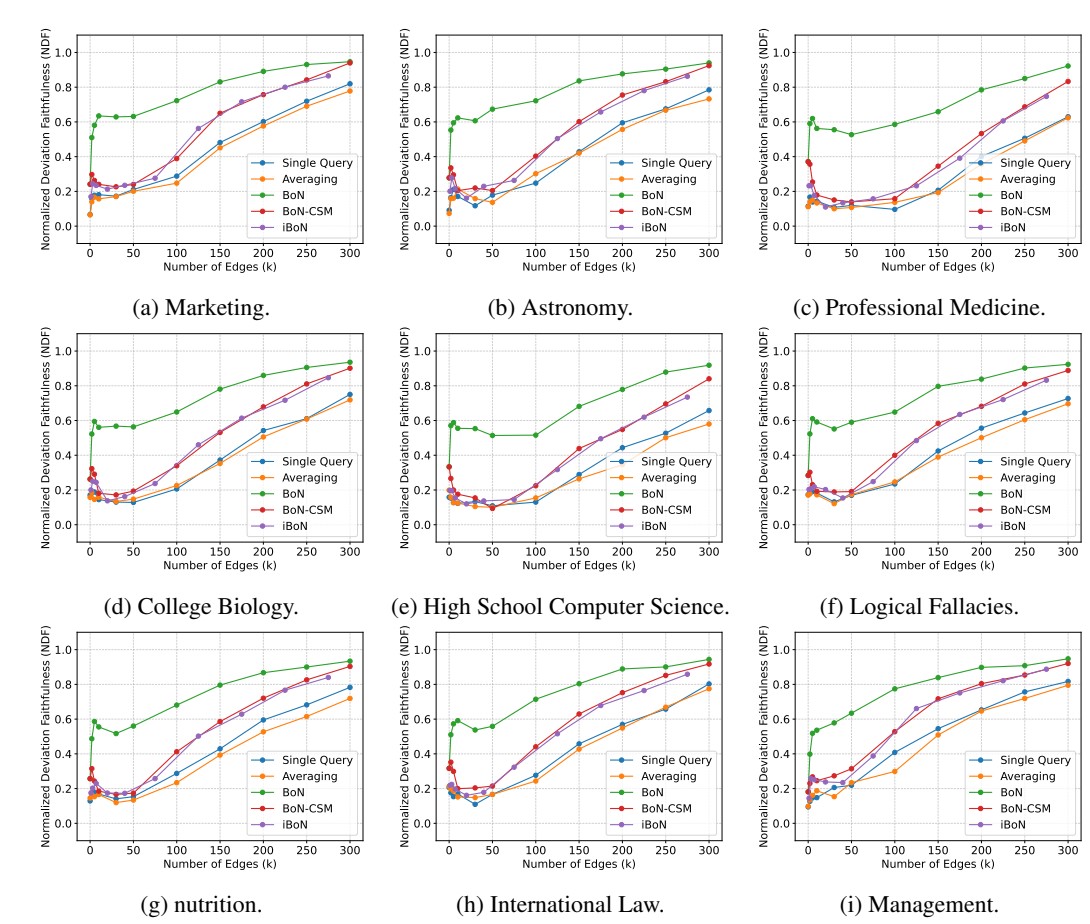

Figure A13: Complete results of BoN sampling for query circuit discovery on nine MMLU categories.

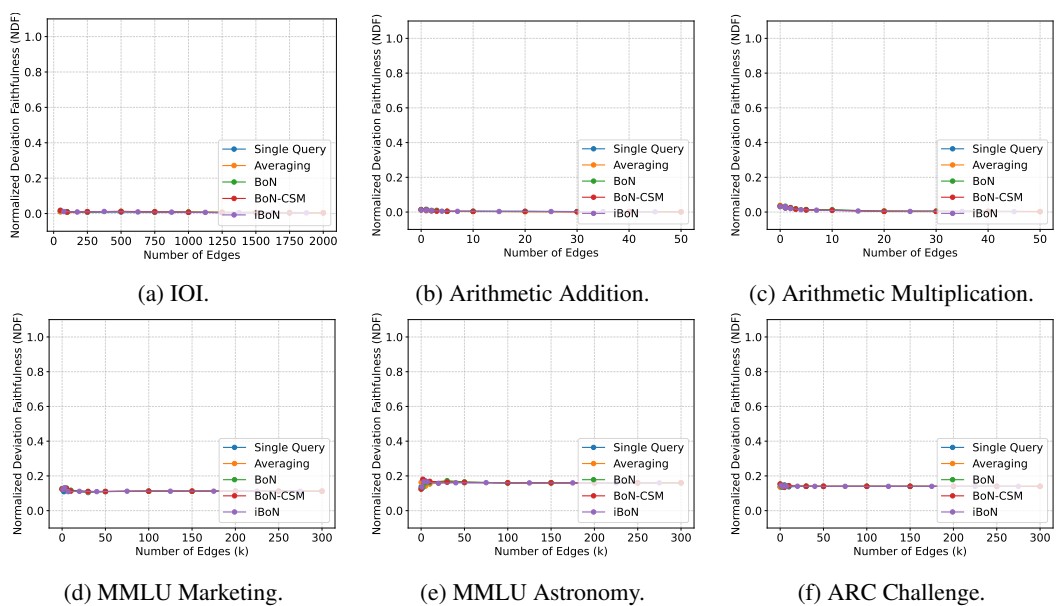

Figure A14: NDF scores of complement circuits of the discovered query circuits.

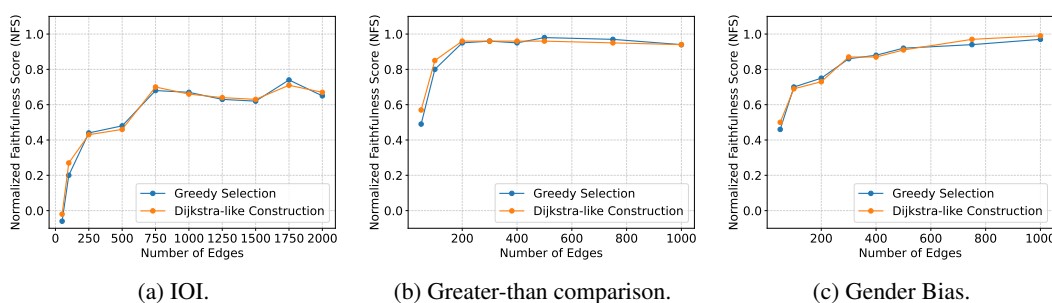

(a) IOI.      (b) Greater-than comparison.      (c) Gender Bias.

Figure A15: Performance comparisons between greedy selection and Dijkstra-like iterative construction for forming a circuit. The two methods achieve similar results on all three tested datasets.

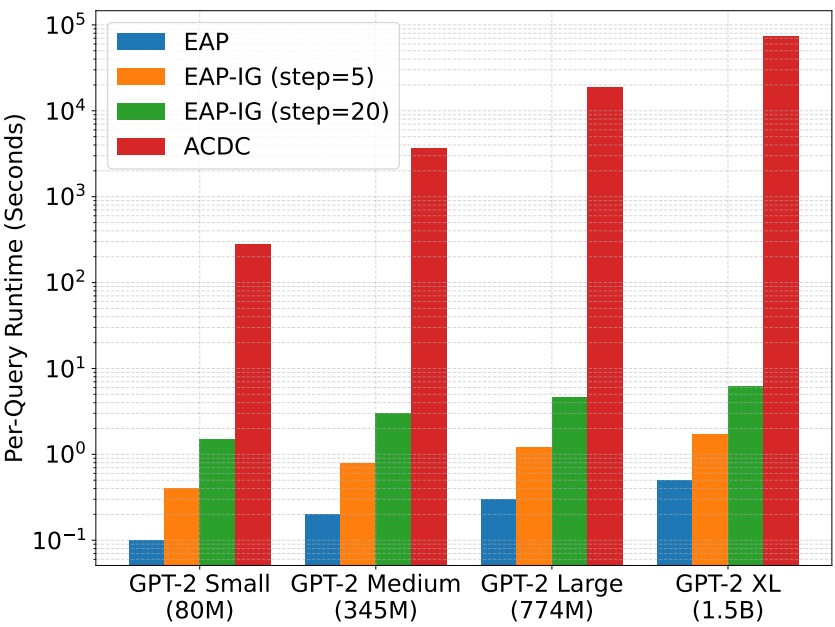

Figure A16: Per-query edge scoring runtime of activation patching (ACDC) versus attribution patching (EAP and EAP-IG) methods. The dataset is IOI. Runtime of ACDC easily grows to hours.

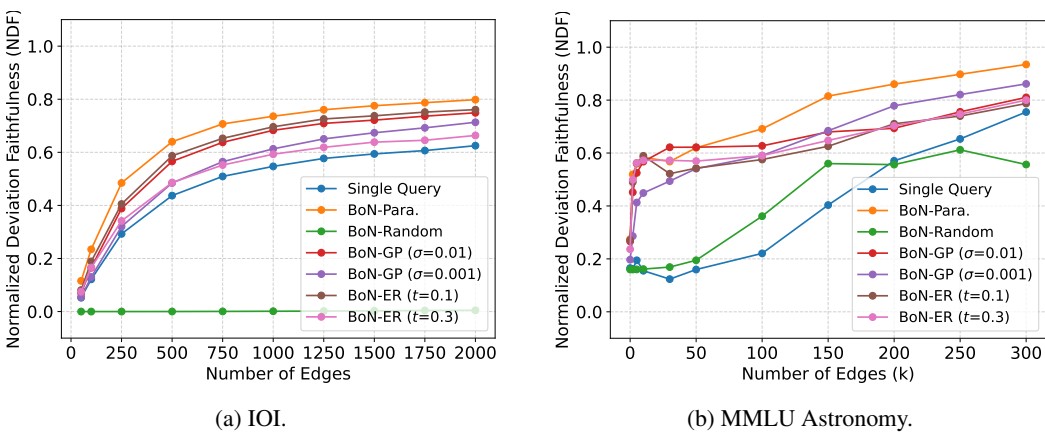

(a) IOI.                 (b) MMLU Astronomy.

Figure A17: Query circuit discovery results for additional BoN variants (BoN-Random, BoN-GP, and BoN-ER), along with BoN by paraphrases (BoN-Para.) introduced in the main paper.

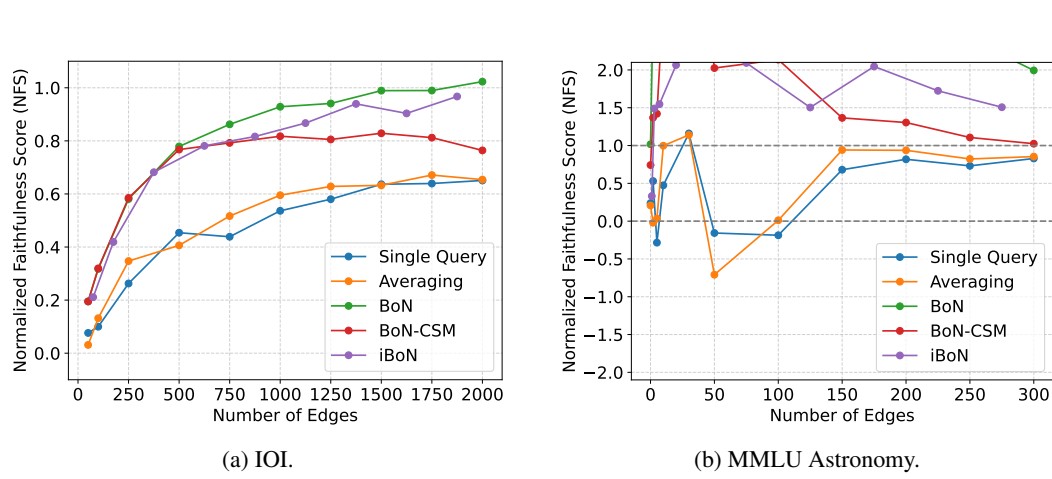

(a) IOI.

(b) MMLU Astronomy.

Figure A18: Query circuit discovery on the full IOI and MMLU Astronomy datasets, evaluated using NFS as in most prior studies of capability circuits. On MMLU, however, NFS fails to provide a stable and reliable evaluation of query circuits and cannot effectively track discovery progress as circuit size increases.

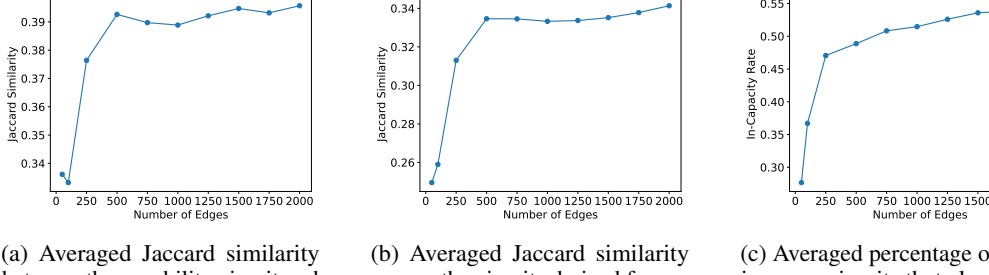

(a) Averaged Jaccard similarity between the capability circuit and query circuits.

(b) Averaged Jaccard similarity among the circuits derived from a query and its paraphrases.

(c) Averaged percentage of edges in query circuits that also appear in the capability circuit.

Figure A19: Analysis on edge overlap.

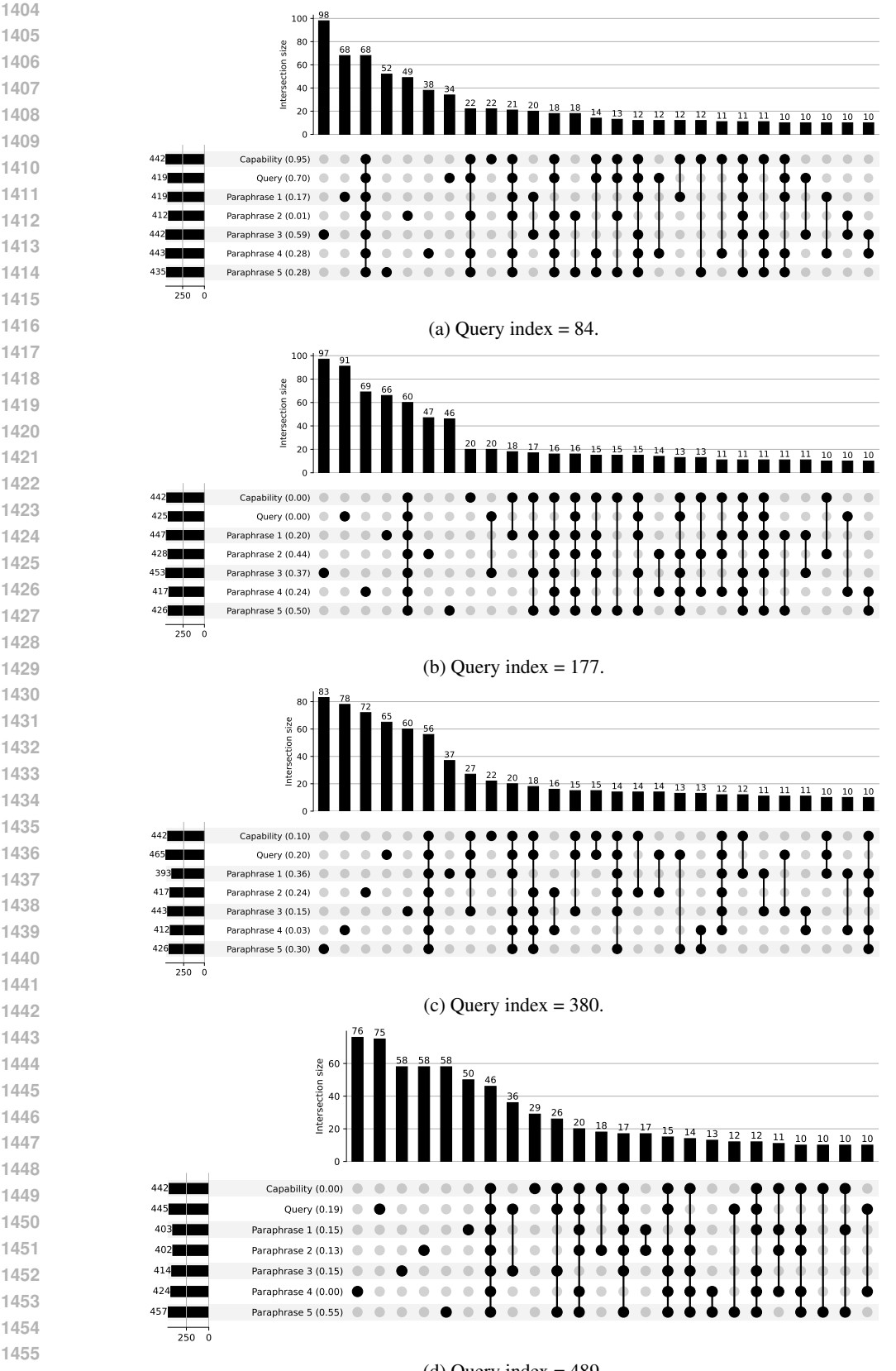

(a) Query index = 84.

(b) Query index = 177.

(c) Query index = 380.

(d) Query index = 489.

Figure A20: Upset plots of four additional randomly selected queries other than the one in Figure 9.

## C JOINT DISCUSSION OF NFS, NDF, AND CMD

This section discusses the relations between NFS, NDF, and CMD metrics. CMD, introduced by the MIB benchmark (Mueller et al., 2025), quantifies how well a capability circuit discovery method identifies circuits that approximate the original model's performance on a given capability. MIB defines the faithfulness of a circuit by NFS.

Let $f(\cdot) : M \to C_k$ denote a circuit discovery method, where $C^k$ is a circuit with $100 \times k$ percentage of the edges of the original LLM $M$. The CMD score of a discovery method is

$$CMD(f) := \int_0^1 \left| 1 - NFS(C^k) \right| dk = \int_0^1 \left| \frac{L(M(D)) - L(C^k(D))}{L(M(D)) - L(M(D'))} \right| dk. \quad (6)$$

A lower CMD score indicates better performance. In practice, the integral is approximated via a Riemann sum, i.e., by evaluating a series of circuits with varying edge budgets. More circuits denote a more precise evaluation. CMD incentivizes each $C^k$ to match the original model's performance and is symmetric with respect to the model performance.

To compare Equation 6 with our NDF metric (Equation 5), we rewrite the NDF of a query circuit $C_q$ as

$$NDF(C_q) = 1 - \min\left( \left| \frac{L(M(q)) - L(C_q(q))}{L(M(q)) - L(M(q'))} \right|, 1 \right) = 1 - \min(|1 - NFS(C_q)|, 1). \quad (7)$$

Thus, mathematically, NDF is to apply the clipping and reversal to the integrand of CMD. With this simple transformation and use as the new definition of circuit faithfulness, we can (1) easily track the discovery progress as the circuit size grows and (2) evaluate the performance of a query circuit discovery method by examining the Pareto frontier, analogous to previous studies in capability circuit discovery (Syed et al., 2024; Hanna et al., 2024; Zhang et al., 2025; Conmy et al., 2023).

## D LIMITATIONS AND FUTURE WORK

First, this work does not resolve the fundamental limitation of using indirect effects as edge scores: the neglect of combinatorial interactions among edges. While fully accounting for such interactions is NP-hard, we believe that empirical and theoretical advances in mechanistic interpretability will enable more efficient estimation of these effects, leading to improved circuit discovery methods.

Second, like all existing circuit discovery methods, this work focuses on queries whose outputs are single tokens, such as option IDs (e.g., "A") in MCQs. This limitation arises because attributing components across edges and forward passes for multi-token generations is complex, and no existing studies have fully addressed this challenge. We believe that efforts to overcome this limitation would be highly valuable for future research in circuit discovery.

Finally, though out of this paper's scope, we believe an automated method for interpreting functionalities and properties of identified edges and nodes is crucial and rewarding. Methods in SAE- and CLT-based circuit discovery provide node interpretations by summarizing each neuron's highly activated tokens. This summarization is tractable since neurons in SAEs and CLTs are highly monosemantic. For circuit discovery within the model, however, raw model components are more polysemantic and thus hard to explain. Previous works explain the discovered circuits by manual investigation, which is not applicable and scalable when the circuit size becomes larger. Nonetheless, recent advances in automated interpretability (Foote et al., 2023; Shaham et al., 2024; Bricken et al., 2025), which automate the human investigation process to study the internals of the LLM, shed light on this issue. We believe streamlining the query circuit discovery and automated interpretability methods is a promising future direction for scalable and faithful decision explanations of LLMs.

