# OpenReview forum: "Query Circuits: Explaining How Language Models Answer User Prompts"
_ICLR.cc/2026/Conference — Submitted to ICLR 2026_

### Official Review · Reviewer_1YXG · 2025-10-29

**Soundness:** 1
**Presentation:** 3
**Contribution:** 2
**Rating:** 2
**Confidence:** 3

**Summary:**

This paper investigates the task of finding input-specific circuits (subgraphs of model components) that explain a model's behavior on a given prompt. The paper first identifies issues with applying existing methods for finding and evaluating dataset-specific circuits, and then proposes a new evaluation metric, and a new circuit finding method, which is based on generating paraphrases of the given query and either aggregating the circuits from different paraphrases. Experiments on four datasets indicate that the proposed methods outperform baselines across sparsity levels.

**Strengths:**

- This paper addresses an interesting problem, explaining how LLMs make decisions for a given input.

- The paper is written clearly and is relatively systematic. It provides convincing evidence that existing circuit finding methods perform poorly in this setting, and introduces a new metric along with a new method that empirically improves performance.

- The best-of-N method seems sensible to me--it increases the set of prompts used for circuit finding, which could reduce noise. This could also mitigate against some issues with query-level circuit finding I discussed in the Weaknesses section, where a circuit could "hard-code" the model's output for a given query. This method can be thought of as creating a small "capability" dataset composed of paraphrases.

- The experiments are thorough, covering four datasets and several different LMs. The results show that the proposed best-of-N methods outperform the single-query baseline, and the results generally improve with increasing N and with more edges.

- The paper introduces two variants of the best-of-N method, which are faster to run.

- The paper is generally clearly written and I found the presentation to be effective.

**Weaknesses:**

_Soundness_

- I think there is a conceptual issue with query-level circuits: if the objective is just to recover the output of model $M$ on a specific input $q$, there is nothing to prevent the method from finding a "constant" circuit that just always outputs $M(q)$, without reflecting the model's underlying computations.

- Similarly, it is unclear to me that NDF (eq. 5) actually captures faithfulness. For the reason mentioned above, it is possible for some query circuit $C_q(q)$ to exactly match the model's output $M(q)$, without being faithful to the model. Specifically, $C_q$ could effectively "hard-code" the model's output, but be invariant to the input. I believe the definition of faithfulness needs to be grounded in some kind of counter-factual notion of faithfulness. See for example [1] for relevant discussion.

- Section 3.3.1 argues that NFS is unstable for MMLU by showing that the score has high variance across number of edges. But this variance could be due to the circuit finding method rather than the metric. This experiment seems to be with query-level circuit finding, rather than task level circuit finding--it seems likely that the variance arises because the method finds different circuits at different N, not because of problems with the metric.

- For this reason, I think it would be very informative to report the NFS score on the paraphrases generated in section 5. This would help understand if the issues identified in Sec. 3 are due to the metric, EAP-IG, or just applying these tools to a single query.

_Contribution_

- One of the stated contribution of this paper is to propose the task of finding input-specific circuits. There is relevant recent work that also studies input-specific circuits (albeit using transcoder features): [2]. I think this paper would benefit from a more extended discussion of that work. In particular, [2] discusses counter-factual experiments for evaluating whether the prompt-level circuits are faithful, which might be applicable here.

- The paper does not give any illustrations of whether the resulting circuits can actually be used to interpret the model.

_Minor comments_

- The name "Query circuits" might be confusing given that "query" also describes a component in the attention mechanism--for example, see https://transformer-circuits.pub/2025/attention-qk/index.html about key-query circuits. A possible alternative could be something like "prompt-level circuits".

- Typo in the abstract: "For example, a circuit covering only 1.3% of model connections can recover about 60% of performance on *an* MMLU questions." Also somewhat unclear to me--does this mean each one circuit for each MMLU question, with no circuit havings sparsity more than 1.3%?

- There is a relevant prior work [3], which introduces an optimization-based method for finding edge circuits. This method mitigates issue with IE noted in section 3.3.2 (ignoring combinatorial effects among edges).

_References_

[1] Geiger et al., 2025. Causal Abstraction: A Theoretical Foundation for Mechanistic Interpretability.

[2] Ameisen et al., 2025. Circuit Tracing: Revealing Computational Graphs in Language Models.

[3] Bhaskar et al., 2024. Finding transformer circuits with edge pruning.

**Summary:** I think the paper proposes a sensible method for an interesting problem, and the paper is generally thorough and clearly written. However, I have a number of doubts about the soundness of the results, given the possibility that query-level circuits can "hard-code" a response; and I feel the paper is missing a number of important baselines (like reporting NFS on paraphrased queries). I am open to increasing my score if the authors can add some discussion of how to address the soundness concerns; add more discussion of the connection to other work on prompt-level circuit finding; and add some of the results I mentioned above.

**Questions:**

- In Fig 2a., did you try measuring NFS for capability circuits for MMLU, rather than query circuits? This could reveal if the variance is because NFS is unreliable on MMLU (as claimed in section 3.3.1), or because query circuits have high variance.

- A natural baseline to compare with Best-of-N sampling is to simply generate $p$ paraphrases of query $q$ and then run EAP-IG with dataset $D = \{q_1, \ldots, q_p\}$. This method might be simpler to implement and more general than the methods presented here, which involve interpolating between multiple potentially large score matrices. Is this the "averaging" baseline in Figure 6? It is not clear to me from the description in section 6.1.


- For best-of-N sampling, do you also permute the order of the answer choices? Do you also paraphrase the possible answers? I think these changes could both help to reduce the likelihood that the circuit hard-codes an answer (although this will still be a possibility).


- Are the results in figure 6 averaged over all of the queries in the dataset? I think it would be helpful to report some of these experimental details in section 6.1 and Figure 6: what is N for best-of-N, and how many examples are in each dataset.


- Have you conducted any analysis into the relationship between query-level circuits for a given task (e.g. MMLU marketing)? For example, for a given task, how sparse is the union of query circuits? How much do the different circuits overlap?

- Similarly, have you investigated the relationship between query-level circuits (e.g. for IOI) and task-level circuits? Are task-level circuits simply the union of query-level circuits? If the circuits are disjoint, this would be reason to suspect that the query-level circuits might be spurious.

- Could this method be applied to circuits composed of SAE features, rather than coarse-grained model components, as in [1]?



[1] Marks et al., 2024. Sparse feature circuits: Discovering and editing interpretable causal graphs in language models

---

> ### Author Response · Authors · 2025-11-21
>
> Thank you for your constructive suggestions. We would like to address your questions and concerns as follows (they are rearranged a bit to enable a smoother response). We have uploaded the revised manuscript. https://anonymous.4open.science/r/iclr2026-query-circuit-rebuttal contains all additional experiments required by reviewers. Official Response to Shared Questions and Concerns (i)–(iii) contains responses to concerns and questions raised by multiple reviewers.
>
> We would like to kindly remind you that the Appendix section in our original submission has solved some of your concerns and questions, while we still find them helpful for us to refine and enhance the paper—thank you.
>
> **1. I think there is a conceptual issue with query-level circuits: if the objective is just to recover the output of model $M$ on a specific input $q$, there is nothing to prevent the method from finding a "constant" circuit that just always outputs $M(q)$, without reflecting the model's underlying computations.**
>
> Please refer to Official Response to Shared Questions and Concerns (i)-(ii).
>
> **2. It is unclear to me that NDF (eq. 5) actually captures faithfulness...*[SKIP]***
>
> Many circuit discovery studies, including ours, have incorporated counterfactual tests. Let $C$ be a capability or query circuit in model $M$. Following prior work (e.g., Figure 3 in [1]), we evaluate the *complement circuit* $C^c \coloneqq M \setminus C$ to perform a *necessity test*: if removing $C$ breaks the model’s behavior, then $C$ is necessary, as shown in Appendix B.4 and Figure A14. Conversely, the *sufficiency test* evaluates the faithfulness of $C$ itself, as shown in Figure 6.
>
> The logic is, if a circuit $C$ indeed contains the model $M$’s critical information of parsing the input and generating the response, ablating it should break the model’s performance on the corresponding query/task. That is, the faithfulness of the complement circuit $C^c$ should be low, as shown in Figure A14 of our paper (and prior studies also have similar conclusions).
>
> In sum, a faithfulness measure (NFS or NDF) alone is indeed not enough to confidently distinguish the genuine circuit from a trivial one, irrelevant to $M$’s underlying computation. Nonetheless, being backed up by the experiment on the complement circuit helps solve the concern.
>
> We’ve merged the above discussion to lines 950-966 in Appendix B.4. Please have a check.
>
> **3. Are the results in figure 6 averaged over all of the queries in the dataset? I think it would be helpful to report some of these experimental details in section 6.1 and Figure 6: what is N for best-of-N, and how many examples are in each dataset.**
>
> Yes. We had a much more detailed experimental setup in Appendix A, as mentioned in Section 6.1. However, we find that the description of “the results in figure 6 averaged over all of the queries in the dataset” is indeed missing—we have added it to lines 424 and 863. Thank you for pointing this out.
>
> We didn’t assign a specific meaning to the *N* of the BoN. It simply means selecting the best circuit among paraphrases and the original query. The number of paraphrases is p=9. In our body text, *N* is used to denote a circuit's number of edges. Please let us know if you think this is over confusing.
>
> **4. A natural baseline to compare with Best-of-N sampling is to simply generate p paraphrases of query q and then run EAP-IG with dataset D=q1,...,qp. *[SKIP]* Is this the "averaging" baseline in Figure 6? It is not clear to me from the description in section 6.1.**
>
> Yes, the “Averaging” baseline in Figure 6 is exactly this method. It is the "baseline (ii)" described in Section 6.1 (lines 429-430). We've added descriptions to lines 462-463 to make it clearer.
>
> [1] Sparse Feature Circuits: Discovering and Editing Interpretable Causal Graphs in Language Models (ICLR'25)

---

> ### Author Response · Authors · 2025-11-21
>
> **5. Section 3.3.1 argues that NFS is unstable for MMLU by showing that the score has high variance across number of edges. But this variance could be due to the circuit finding method rather than the metric. This experiment seems to be with query-level circuit finding, rather than task level circuit finding--it seems likely that the variance arises because the method finds different circuits at different N, not because of problems with the metric.**
>
> **For this reason, I think it would be very informative to report the NFS score on the paraphrases generated in section 5. This would help understand if the issues identified in Sec. 3 are due to the metric, EAP-IG, or just applying these tools to a single query.**
>
> **Q: In Fig 2a., did you try measuring NFS for capability circuits for MMLU, rather than query circuits? This could reveal if the variance is because NFS is unreliable on MMLU (as claimed in section 3.3.1), or because query circuits have high variance.**
>
> Yes, we’ve done similar experiments in Appendix B.8. We had two observations:
> (a) The NFS of the single query circuit on MMLU is unstable (Figures 2a and 3).
>
> (b) The NFS of circuits by the “averaging” baseline on MMLU is unstable (Figure A18 (b) in Appendix B.8; the curve is averaged over all samples in MMLU Astronomy).
>
> The "Averaging" baseline is like finding the “capability circuits for an MMLU query”. In these settings, we still find NFS to be suboptimal. Besides, it is unlikely to be the issue of EAP-IG, as EAP-IG pairing with NDF shows stable progress.
>
> In the folder “rephrase_whole_queries/”, we show NDF and NFS on query circuits discovered by the original query and its paraphrases (similar to Figure 4, but contains an NFS version).  NDF still produces a stabler evaluation.
>
> We noticed that you mentioned “capability circuits for MMLU” in your questions. In case of any confusion, we would like to clarify that there is generally no “capability circuit for MMLU”. This is because: “capability circuit of MMLU” means the one that can recover model performance on the entire dataset. However, even within a category, each query is quite semantically and linguistically different from the other; it is counterintuitive that there exists a sparse subnetwork in the model capable of answering all these queries.
>
> **6. One of the stated contribution of this paper is to propose the task of finding input-specific circuits. There is relevant recent work that also studies input-specific circuits (albeit using transcoder features): [2]. I think this paper would benefit from a more extended discussion of that work. In particular, [2] discusses counter-factual experiments for evaluating whether the prompt-level circuits are faithful, which might be applicable here.**
>
> We would like to clarify that our stated contribution is input-specific circuits “directly inside” the model, not on replacement models (e.g., transcoders) trained to approximate it. We added and modified some sentences in lines 43-53 to make our niche clearer.
>
> For counter-factual experiments in [2], we assume you referred to their Section 3.5 and Figure 10, while that section is more for qualitative examples (manual investigation). Some papers, like feature circuits [1], have quantitative counter-factual experiments, which we have also conducted in Appendix B.4.
>
> However, it is still a good reference, and we think it is related to your request on “illustrations of whether the resulting circuits can actually be used to interpret the model,”—we will add such examples in the camera-ready version, as discussed and scheduled in Official Response to Shared Questions and Concerns (iii).
>
> **7. The name "Query circuits" might be confusing given that "query" also describes a component in the attention mechanism--for example, see https://transformer-circuits.pub/2025/attention-qk/index.html about key-query circuits. A possible alternative could be something like "prompt-level circuits".**
>
> Thank you for this good suggestion. We will consider it and make a corresponding revision through the paper in the camera-ready version. We currently use the word "query circuits," while we also mention it as "prompt-level explanations" in line 69.
>
> [1] Sparse Feature Circuits: Discovering and Editing Interpretable Causal Graphs in Language Models (ICLR'25)
> [2] Circuit Tracing: Revealing Computational Graphs in Language Models (Anthropic, https://transformer-circuits.pub/2025/attribution-graphs/methods.html)

---

> ### Author Response · Authors · 2025-11-21
>
> **8. Typo in the abstract: "For example, a circuit covering only 1.3% of model connections can recover about 60% of performance on an MMLU questions." Also somewhat unclear to me--does this mean each one circuit for each MMLU question, with no circuit havings sparsity more than 1.3%?**
>
> We have fixed the typo “s” in the uploaded manuscript. Thank you.
>
> The original abstract missed the word “on average,” as has been used in line 84. We have added it back (line 24). The correct interpretation is: *some query circuits need few edges to recover 60% model performance on their corresponding queries; some need more edges. On average, they need 1.3% of the model's edges to achieve 60% performance.*
>
> **9. There is a relevant prior work [3], which introduces an optimization-based method for finding edge circuits. This method mitigates issue with IE noted in section 3.3.2 (ignoring combinatorial effects among edges).**
>
> Optimization-based methods only work in the setting of capability circuit; they are incompatible for finding the circuit of a single query. A clear issue is that optimization-based methods require a training set to optimize edge and node masks—when the training set contains only one query, the optimization process quickly falls into a local minima and does not properly prune edges. The resulting circuits are extremely sparse, thus malfunctioning (NFS=NDF=0 on the same query).
>
> The above argument is supported by our experimental results in the folder “Edge_Pruning_on_IOI/” at https://anonymous.4open.science/r/iclr2026-query-circuit-rebuttal. Under “Edge_Pruning_on_IOI/”, the “replication/” folder contains reproduction of Edge Pruning [3], aligning with numbers reported in the original paper; Under the “single query setting/“ folder, the “query circuit evaluation result/” folder contains 10 query circuits’ performance discovered by Edge-Pruning.; the “loss plot/” folder contains each query circuit's training loss (two antagonistic loss functions). Among the 10 tested queries, their training losses do converge, while the discovered query circuits are all unfaithful (NDF=NFS=0).
>
> **10. For best-of-N sampling, do you also permute the order of the answer choices? Do you also paraphrase the possible answers? I think these changes could both help to reduce the likelihood that the circuit hard-codes an answer (although this will still be a possibility).**
>
> We have tried paraphrasing the entire query, including answer choices. Results are similar to rephrasing question stems only, as shown in figures under the folder "rephrase_whole_queries/".
>
> We didn’t adopt this setting in our submission since altering question choices may invoke concerns about semantic shift—for example, in some questions, answer choices are professional terms, and changing them to any other words may break the intention of the question.
>
> For the concern of “likelihood that the circuit hard-codes an answer,” please refer to Official Response to Shared Questions and Concerns (ii).
>
> **11. Could this method be applied to circuits composed of SAE features, rather than coarse-grained model components, as in [1]?**
>
> Yes, it can. Feature circuits [1] are compatible with our work. This is also mentioned in Official Response to Shared Questions and Concerns (iii). Here is a detailed response:
>
> Feature circuits apply IE to find circuits on replacement models (SAEs). In contrast, our main technical argument is that IE-based circuits, when applied to a single query, often fall into a “local minima” due to IE’s inherent limitation (which may cause a mirage that “circuit for a single query is infeasible”)—while simply rephrasing the query can, in many times, get out of the local minima and find a decent query circuit via perturbing the edge score matrix. Hence, when you are using [1] to find the feature circuit of a single query, our paper suggests that you do some perturbations on the query, and use NDF over NFS (and our paper explained why these practices are good).
>
> If you have any remaining concerns or questions not covered above, please feel free to let us know—we are happy to address them.
>
> [1] Sparse Feature Circuits: Discovering and Editing Interpretable Causal Graphs in Language Models (ICLR'25)
> [3] Finding Transformer Circuits with Edge Pruning (NeurIPS'24)

---

### Official Review · Reviewer_3hrb · 2025-10-31

**Soundness:** 2
**Presentation:** 2
**Contribution:** 2
**Rating:** 4
**Confidence:** 3

**Summary:**

This paper attempts to explain a model's output on specific inputs through network connections, i.e., circuits. Unlike functional circuit discovery which focuses on general algorithmic capabilities, this method aims to provide local, query-level explanations. To better evaluate the faithfulness of discovered circuits, the paper introduces an improved evaluation metric (NDF). Two sampling methods are proposed that use paraphrases to help recover model behavior. On MMLU questions, circuits using only 1.3% of the model's connections can recover approximately 60% of the model's performance. However, the gap between "finding sparse subnetworks that preserve performance" and "explaining how the model actually processes this input" remains substantial.

**Strengths:**

- The paper is well-structured with clear problem formulation. The distinction between capability circuits and query circuits is well-articulated, and the motivation for local explanations is compelling. The proposed metric is a simple and well-motivated fix.
- Using paraphrases to provide semantically-equivalent but slightly perturbed samples is intuitive and straightforward. The "lottery ticket" framing helps conceptualize why this approach works. The results are also strong and consistent across diverse benchmarks.
- The experimental design is thorough. Tests span multiple benchmarks with varying complexity levels, include multiple baselines and ablations.

**Weaknesses:**

- Selecting the "best" circuit through empirical search demonstrates that some set of connections can reproduce outputs effectively. However, it's unclear whether this constitutes finding the query circuit that explains how the model actually processed that specific input, or merely identifies one among many possible circuits that happen to work.
- Functional circuit work (e.g., for addition or IOI) typically provides detailed mechanistic interpretations showing how circuits implement specific computations. This paper doesn't establish such demonstrations for query circuits, no analysis of what roles different nodes/edges play or why discovered circuits work.
- The general query circuit discovery in this paper bears a strong assumption that there exists a meaningful general query circuit for each input that explains the model's reasoning. However, Is there actually a unique, identifiable circuit the model uses for each query? Or are there multiple valid circuits that could produce the same output? Can we distinguish a true "query circuit" from an arbitrary subset of connections that happens to preserve performance?
- While the sampling method successfully identifies subnetworks that govern generation abilities, it doesn't provide explanations for how or why they work. The method could succeed even if the underlying theoretical motivation is incorrect, it's fundamentally a search procedure over candidates. Alternatively, it is difficult to determine whether the paper has found the actual query circuit used by the model or a subnetwork storing important knowledge.

**Questions:**

Please see weakness.

---

> ### Author Response · Authors · 2025-11-21
>
> Thank you for the insightful feedback and comments. We would like to discuss your concerns and how we have addressed them. We have uploaded the revised manuscript. https://anonymous.4open.science/r/iclr2026-query-circuit-rebuttal contains all additional experiments required by reviewers. Official Response to Shared Questions and Concerns (i)–(iii) contains responses to concerns and questions raised by multiple reviewers.
>
> **1. It's unclear whether BoN constitutes finding the query circuit that explains how the model actually processed that specific input, or merely identifies one among many possible circuits that happen to work.**
>
> Please refer to Official Response to Shared Questions and Concerns (i) and (ii) which contain thorough analysis to solve this concern.
>
> **2. Is there actually a unique, identifiable circuit the model uses for each query? Or are there multiple valid circuits that could produce the same output? Can we distinguish a true "query circuit" from an arbitrary subset of connections that happens to preserve performance?**
>
> Please refer to Official Response to Shared Questions and Concerns (i)-(ii). I believe arguments there answer these questions well. In short:
>
> (1) There's no guarantee of the "uniqueness" of the circuit. There are potentially multiple valid sub-circuits that the model simultaneously relies on, and setting the edge budget somehow mediates the selection of them, as argued in Official Response to Shared Questions and Concerns (ii).
>
> (2) Query circuit is unlikely to be just an arbitrary subset of connections, as argued in Official Response to Shared Questions and Concerns (i).
>
> **3. Functional circuit work (e.g., for addition or IOI) typically provides detailed mechanistic interpretations showing how circuits implement specific computations. This paper doesn't establish such demonstrations for query circuits, no analysis of what roles different nodes/edges play or why discovered circuits work.**
>
> Please refer to Official Response to Shared Questions and Concerns (iii). In short, we will incorporate such demonstrations for MMLU questions in the camera-ready version. The experiment is in progress now.
>
> Notably, although early functional circuit work indeed provides interpretations by manual investigation on all model components, later studies [1-7] focus more on automating the process of circuit identification (i.e., circuit discovery). The two groups of studies are complement to each other. One can surely conduct manual investigation on automatically identified sparse circuits, which should be easier than traversing all model components.
>
> Studies [1-7] focusing on in-place circuit discovery mostly report circuit diagrams only. They focus more on studying the properties (e.g., faithfulness and inference time) of proposed discovery methods. This paper aligns with their practice, and we also show circuit diagrams in Table A5 in the revised manuscript.
>
> **4. The paper doesn't provide explanations for how or why the sampling method works. The method could succeed even if the underlying theoretical motivation is incorrect, it's fundamentally a search procedure over candidates. Alternatively, it is difficult to determine whether the paper has found the actual query circuit used by the model or a subnetwork storing important knowledge.**
>
> Please refer to Official Response to Shared Questions and Concerns (i) and (ii). In short, we show that BoN functions as a search procedure over non-arbitrary candidate circuits: circuits discovered from different paraphrases consistently preserve a shared sub-circuit. Plus, calculations of IEs tend to select edges underlying the model's critical computation, supported by our necessity test in Figure A14. These results indicate that BoN is capturing computation that the model genuinely relies on, rather than stumbling upon unrelated subnetworks.
>
> If you have any remaining concerns or questions not covered above, please feel free to let us know—we are happy to address them.
>
> [1] Have Faith in Faithfulness: Going Beyond Circuit Overlap When Finding Model Mechanisms (COLM'24)
> [2] Towards Automated Circuit Discovery for Mechanistic Interpretability (NeurIPS'23)
> [3] Finding Transformer Circuits with Edge Pruning (NeurIPS'24)
> [4] Sheaf Discovery with Joint Computation Graph Pruning and Flexible Granularity (EMNLP'25)
> [5] Position-aware Automatic Circuit Discovery (ACL'25)
> [6] Information Flow Routes: Automatically Interpreting Language Models at Scale (EMNLP'24)
> [7] Circuit Stability Characterizes Language Model Generalization (ACL'25)

---

> > ### Comment · Reviewer_3hrb · 2025-11-26
> >
> > I thank the authors for the detailed response and the additional experiments regarding IOI and circuit overlap. While these results confirm that the discovered subgraphs are robust and functionally necessary, the critical concern regarding the definition of a "query circuit" remains.
> >
> > Demonstrating that a subgraph is critical for performance (via ablation) does not automatically guarantee its explainability. Furthermore, the significant overlap with capability circuits suggests the method isolates the general machinery required for the model to function on the task, rather than the specific reasoning path unique to the query. While providing concrete qualitative usage examples could help validate the definition of a "query circuit," the current absence of such evidence makes it difficult to distinguish the discovered subgraphs from general task-relevant activations.
> >
> > In summary, my concerns remain, and I will maintain my score.

---

> ### Author Response · Authors · 2025-11-26
>
> Thank you for the response. We would like to further address your concern. Please let us know if our clarification does not solve the main issue.
>
> *Explainability* has different granularities. **Circuits alone guarantee the explanation of how the model actually processes the query using a particular small subset of its components, rather than layer-by-layer stacking of all representations,** shown via sufficiency and necessity tests in our paper and prior circuit discovery works. The sufficiency and necessity tests make query circuits well-defined. Indeed, to explain the role of each component by human-readable concepts (finest granularity), circuits have to pair with additional techniques like SAEs; while our work is technically compatible with SAEs (as discussed in point 11 in our response to Reviewer 1YXG).
>
> For the significant overlap with the capability circuit (on the IOI experiment), we want to clarify that **this is expected and should be the case to support our argument of non-triviality of query circuits (as mentioned by Reviewer 1YXG and 7mAG; please refer to the 6th point of the "Questions" section in Reviewer 1YXG's initial review)**. Note that the capability circuit is the one expected to handle all IOI queries (paraphrase of the other), and the query circuit is the one expected to handle a specific IOI query. They should share a similar reasoning path (always-in subgraph) relevant to general task-relevant activations; otherwise, we may suspect query-level circuits to be spurious. This experiment is a special case to further verify the non-spuriousness of query circuits, as a non-toy dataset in the real-world generally does not have a capability circuit.
>
> Concretely, **the query circuit is well-defined in non-toy datasets since capability circuits generally do not exist there.** On a non-toy dataset like MMLU, one may only find the circuit for each query. Each query is very different from the other, and so there's generally no many "general task-relevant activations"  For example, we won't expect there to be a sparse subnetwork in Llama-3.2-1B to recover its performance on the whole MMLU dataset (i.e., the capability circuit of MMLU)—intuitively, the model should have used its full components to tackle this dataset, as different queries in MMLU require different reasoning and factual recalls.
>
> We hope that your concern can be mitigated through this clarification, and we look forward to your response. Thank you again for your valuable feedback.

---

### Official Review · Reviewer_UiWe · 2025-11-01

**Soundness:** 3
**Presentation:** 3
**Contribution:** 3
**Rating:** 8
**Confidence:** 4

**Summary:**

This paper proposes a new concept, query circuits, which recover LLM behavior by mimicking the information flow through the network. The authors measure the efficacy of their query circuits by their Normalized Deviation Faithfulness, which looks at the model's recovery on corrupted queries. Finally, the authors propose BoN for circuit discovery, and find that it does much better than previous algorithms for circuit discovery.

**Strengths:**

- The paper proposes a natural direction for circuit discovery (looking at local information flow as opposed to global algorithms) and also develops a solid method (BoN) that scales with more compute.
- The paper derives a more natural method for BoN and studies many of its derivatives.
- The results are promising and suggest that query circuits are recovering meaningful aspects of information flow in the model.

**Weaknesses:**

- The discovered circuits are able to recover performance in the network, but they are likely not as interpretable and informative as more global capability circuits. While I don't think this is an inherent limitation, as there are many novel applications that can be explored by having good query circuits, there does appear to be a fundamental limitation in "understanding" LLM behavior if one only uses query circuits.

**Questions:**

- Could there be more discussion of how these query circuits might be used in model interpretability? Identifying harmful query circuits (e.g., for jailbreaks) and pruning them + adjusting model behavior would be interesting.

---

> ### Author Response · Authors · 2025-11-21
>
> Thank you for appreciating our work. Below, we discuss your questions and concerns. We have uploaded the revised manuscript. https://anonymous.4open.science/r/iclr2026-query-circuit-rebuttal contains all additional experiments required by reviewers. Official Response to Shared Questions and Concerns (i)–(iii) contains responses to concerns and questions raised by multiple reviewers.
>
> **1. The discovered circuits are able to recover performance in the network, but they are likely not as interpretable and informative as more global capability circuits.**
>
> We partially agree with this point:
>
> First of all, global capability circuits are interpretable and informative because they are obtained from manual investigation, which is tractable only when on small models with a low number of edges and has only one circuit to dissect.
>
> On the other hand, in query circuit discovery, it’s intractable to manually investigate the roles of nodes for the circuit of every incoming query. We believe a better practice is to pair with SAE features [1] or automated interpretability [2]. However, SAE features and/or automated interpretability with prompt-level circuits might be another long topic, so we decided to let them be in future work, focusing on consolidating the arguments of the proposed new avenue (i.e., query circuit discovery).
>
> For automated interpretability [2], an emerging paradigm for understanding LLM behaviors, their systems (agents) are usually equipped with a set of “tools” (e.g., getting highly-activated neurons of an LLM response) to plan and conduct interpretability research on their own. In this context, query circuit discovery may also bear the value of serving as a new category of tools for the system to make it more capable.
>
> **2. More discussion of how these query circuits might be used in model interpretability?**
> Please refer to Official Response to Shared Questions and Concerns (iii).
>
> If you have any remaining concerns or questions not covered above, please feel free to let us know—we are happy to address them.
>
> [1] Sparse Feature Circuits: Discovering and Editing Interpretable Causal Graphs in Language Models (ICLR'25)
> [2] Building and evaluating alignment auditing agents (Anthropic, https://alignment.anthropic.com/2025/automated-auditing/)

---

### Official Review · Reviewer_7mAG · 2025-11-01

**Soundness:** 3
**Presentation:** 3
**Contribution:** 2
**Rating:** 6
**Confidence:** 3

**Summary:**

The paper uses mechanistic interpretability’s circuit discovery for LLMs at the single-query (atomic) level, instead of the frequent capability/dataset level. To analyze these queries/prompts it uses edge-attribution patching methods (EAP,  EAP-IG) on different benchmarks (IOI, arithmetic, MMLU, and ARC). As circuits on the atomic level quickly lead to unstable faithfulness results on symmetric faithfulness (NFS), the paper proposes an updated faithfulness metric (NDF) to evaluate the discovered circuits and also uses multiple samples per query, where it produces different paraphrases of the original query for stabilization.
It uses GPT-2 Small and Llama-3.2-1B-Instruct and shows the discovered circuits recover ~60% of the model’s behavior with only ~1–2% of edges, where the edge-attribution score matrices from several paraphrases share similar patterns, while at the same time small score differences between paraphrases can change them, highlighting the necessity of using several paraphrases.

**Strengths:**

- Analyzing prompts/queries on circuit level is an interesting addition to the interpretability toolbox, highlighting e.g., a model’s in-context learning capabilities.
- Similarly, showing that circuits can also be analyzed on an atomic level opens up new avenue and using sampled paraphrasing is a simple yet effective way to overcome brittleness.
- The paper proposes an updated faithfulness metric tocomparte different prompts/paraphrases per query.
- The approach is used on real-world datasets (beyond toy datasets).

**Weaknesses:**

- The discovered circuits are still to be treated with caution as small shifts in one query paraphrase can highly influence the scores of multiple paraphrases.
- The paper only uses small models, so the generalizability is somewhat limited here.
- The method relies in huge parts on previous work from circuit discovery and as such presents only moderate methodological novelty.

**Questions:**

Can you quantify variance in circuits across paraphrases for each prompt and task? (That would make the brittleness directly visible.)
Do the high-scoring edges across paraphrases cluster in specific layers/heads (i.e. is there a consistent “always-on” subgraph)?

---

> ### Author Response · Authors · 2025-11-21
>
> Thank you for your valuable feedback and comments. We would like to address your questions and concerns below. We have uploaded the revised manuscript. https://anonymous.4open.science/r/iclr2026-query-circuit-rebuttal contains all additional experiments required by reviewers. Official Response to Shared Questions and Concerns (i)–(iii) contains responses to concerns and questions raised by multiple reviewers.
>
> **1. The discovered circuits are still to be treated with caution as small shifts in one query paraphrase can highly influence the scores of multiple paraphrases.**
>
> Could you please elaborate more on this question? Small shifts (did you mean wordings, etc?) in one paraphrase will not affect the scores of others. As shown in Figure 5, each paraphrase generates one circuit, which is evaluated independently.
>
> **2. The paper only uses small models, so the generalizability is somewhat limited here.**
>
> The folder “scaling_to_larger_models/” at https://anonymous.4open.science/r/iclr2026-query-circuit-rebuttal shows the generalizability of our methods and findings on larger models. Specifically, we test GPT2-XL (2235025 edges) on IOI and Llama-3-8B-Instruct (1592881 edges) on MMLU astronomy and arithmetic multiplication; Results are similar to the case of GPT2-Small and Llama-3.2-1B-Instruct—for example, in MMLU astronomy with Llama-3-8B-Instruct, 5000 edges (0.3% of all edges) can recover NDF=0.4; if using vanilla EAP-IG, even 300k edges (18%) cannot achieve that.
>
> The above results are added to lines 473-479 in the uploaded revised manuscript.
>
> **3. The method relies in huge parts on previous work from circuit discovery and as such presents only moderate methodological novelty.**
>
> We partially agree with this point. Circuit discovery is one of the primary ways for constructing the causal graph of the model; it has been a popular and important domain, with Anthropic’s series of work [9], deployed service on Neuronpedia [10], and many other studies [1-8].
>
> However, we do agree that the major contribution of this paper is not new algorithms—BoN is methodologically straightforward. The major contributions are: (1) we identify why prior circuit discovery may fall short in a single query setting. (2) We find that a simple technique (BoN) and NDF can help solve the identified issues. (3) We paved the road for follow-up work on this new problem setting, which we believe is interesting and important.
>
> **4. Quantifying variance in circuits across paraphrases for each prompt and task**
> Please refer to Official Response to Shared Questions and Concerns (i).
>
> If you have any remaining concerns or questions not covered above, please feel free to let us know—we are happy to address them.
>
> [1] Sparse Feature Circuits: Discovering and Editing Interpretable Causal Graphs in Language Models (ICLR'25 oral)
> [2] Have Faith in Faithfulness: Going Beyond Circuit Overlap When Finding Model Mechanisms (COLM'24 oral)
> [3] Towards Automated Circuit Discovery for Mechanistic Interpretability (NeurIPS'23 spotlight)
> [4] Finding Transformer Circuits with Edge Pruning (NeurIPS'24)
> [5] Sheaf Discovery with Joint Computation Graph Pruning and Flexible Granularity (EMNLP'25)
> [6] Position-aware Automatic Circuit Discovery (ACL'25)
> [7] Information Flow Routes: Automatically Interpreting Language Models at Scale (EMNLP'24)
> [8] Circuit Stability Characterizes Language Model Generalization (ACL'25)
> [9] https://transformer-circuits.pub/
> [10] https://www.neuronpedia.org/gemma-2-2b/graph

---

### Author Response · Authors · 2025-11-21
**Official Response to Shared Questions and Concerns (i)**

This response is for concerns and questions shared by more than one reviewer.

**1. More analysis on the relationship between capability and query circuits, and among query circuits**

We use IOI dataset to conduct a thorough analysis, with results in the folder “more_circuit_analysis/” at https://anonymous.4open.science/r/iclr2026-query-circuit-rebuttal. It particularly answers whether BoN sampling (1) merely produces unrelated disjoint circuits where some coincidentally output the correct token, or (2) discovers variants of a common mechanism that preserve a shared set of critical edges (i.e., shared sub-circuit), regardless of how the query is phrased.  We find (2) to be the case.

Specifically, “more_circuit_analysis/” contains four experiments to support (2):

(a) The Jaccard similarity consistently falls in [0.3, 0.4], as shown in *jaccard_similarity_qc_ltop.pdf* and *jaccard_similarity_qq_ltop.pdf* (*qc* means comparison between query and capability circuit; *qq* means among query circuits). This indicates non-trivial edge overlap among these circuits. We also report experiments on scaling to larger IOI circuits (more edge budgets), with Jaccard similarity becoming larger, as shown in *jaccard_similarity_qc_htop.pdf* and *jaccard_similarity_qq_htop.pdf*.

(b) We calculate the percentage of edges in a query circuit that are also in the capability circuit. Roughly, an IOI query circuit with 1000 edges has half of its edges also residing in the capability circuit, as shown in *intersect_ratio_qc_ltop.pdf* and *intersect_ratio_qc_htop.pdf*. This result echoes with (a)'s results.

(c) UpSet plots in folder “edgeNum_{500 or 1000}\_paraNum\_{5 or 10}/” display detailed statistics of edge overlap between the capability circuit, query circuit discovered by the query, and query circuits discovered by that query’s paraphrases. The folder contains results of 10 randomly selected queries (sample\_index=0, 177,...).

(d) In the folder “circuit_plot/”, we plot the query circuits of a randomly selected query and the IOI capability circuit. **Nodes and edges shared among all of them are marked in red, constituting a critical “common sub-circuit”**; green nodes and edges are those unique to the residing circuit.

The above results form a firm argument that BoN samples variants of a common, underlying mechanism preserving a shared set of critical edges. We’ve incorporated them as Section 6.4 and Appendix B.9 in the revised manuscript. Please have a check.

---

> ### Author Response · Authors · 2025-11-22
> **Official Response to Shared Questions and Concerns (ii)**
>
> **2. How do you know the discovered circuit is not a trivial one (e.g., a circuit always producing the “C” token)?**
>
> There are four reasons to jointly support the argument that the automatically discovered in-place circuits (either capability circuit by prior work or query circuit by our work) should be non-trivial ones with respect to the model’s underlying computation:
>
> (a) Early work [1, 2] has shown that on datasets with known, human-annotated ground-truth circuits (e.g., IOI, Greater-than), the circuits identified by IE-based methods closely match the ground truth.
>
> (b) To further support non-triviality, several prior studies [3]—and also our paper—conduct counterfactual experiments by ablating the discovered circuit and measuring the resulting performance degradation. Consistently, ablating the circuit breaks the model’s performance on the original task/query (e.g., Figure 3 of Feature Circuits [3] and Figure A14 of our paper), indicating that the model indeed relies on the discovered circuit to perform the computation.
>
> (c) In-place circuit discovery identifies edges with high influence (IE) on the model’s output. Thus, even if a trivial “constant-value circuit” exists inside the model, if the model does not rely on it to generate the answer, it will have low IE scores and therefore tend not to be selected. While we do not deny the possible existence of trivial circuits, IE-based methods naturally prioritize edges that are important for the actual computation over trivial sub-circuits.
>
> (d) Based on the experimental results in Official Response to Shared Questions and Concerns (i), if the identified query circuit were merely a trivial graph that happens to produce a high logit for the answer token, it would be unlikely to share a subgraph with the IOI capability circuit and with the query circuits derived from different paraphrases.
>
> **Discussion on “uniqueness of circuits” (Reviewer 3hrb):**
> To our knowledge, the circuit-discovery literature does not claim that circuits are unique. Indeed, insights from other domains suggest that models can employ multiple internal pathways when producing outputs [4]. In the context of circuit discovery, this can be understood as follows: as the size of the circuit grows (i.e., as more edges are included), additional implicit pathways may appear. Conceptually, the size of the identified circuit (i.e., the edge budget *N* in our paper) depends on whether one aims to isolate only “the most critical computational pathway” or also include “secondary pathways.”
>
> [1] Have Faith in Faithfulness: Going Beyond Circuit Overlap When Finding Model Mechanisms (COLM'24)
> [2] Towards Automated Circuit Discovery for Mechanistic Interpretability (NeurIPS'23)
> [3] Sparse Feature Circuits: Discovering and Editing Interpretable Causal Graphs in Language Models (ICLR'25)
> [4] Chain-of-Thought Is Not Explainability (alphaXiv)

---

> ### Author Response · Authors · 2025-11-22
> **Official Response to Shared Questions and Concerns (iii)**
>
> **3. Better to give an example of using the resulting circuits to interpret the model.**
> We appreciate this suggestion and fully agree that usage examples would substantially strengthen the presentation. We plan to include such illustrations in the camera-ready version. We did not incorporate them in the initial submission because translating circuit components into human-interpretable concepts is a broad and involved topic—large enough to merit a dedicated paper. For this submission, we focused on our core contributions: introducing the in-place query circuit discovery setting, identifying limitations of prior approaches, and demonstrating the feasibility and promise of the proposed method.
>
> To address this feedback in the camera-ready version, we are planning to generate usage examples through one of two approaches: (1) following prior capability-circuit work (e.g., IOI) by manually inspecting attention patterns and logit attributions within the query circuits for selected MMLU questions; or (2) adapting SAE-based feature-circuit methods to the prompt-level setting, incorporating our insights on prompt perturbations and NDF-based evaluation. This experiment is in progress now.
>
> Finally, we note that most prior in-place circuit discovery studies [1–7] have presented circuit diagrams as their main interpretability artifact, without manually annotating the internal roles of every component. Our work now provides circuit diagrams at the same level of granularity, shown in Table A5 of the revised manuscript.
>
> [1] Have Faith in Faithfulness: Going Beyond Circuit Overlap When Finding Model Mechanisms (COLM'24)
> [2] Towards Automated Circuit Discovery for Mechanistic Interpretability (NeurIPS'23)
> [3] Finding Transformer Circuits with Edge Pruning (NeurIPS'24)
> [4] Sheaf Discovery with Joint Computation Graph Pruning and Flexible Granularity (EMNLP'25)
> [5] Position-aware Automatic Circuit Discovery (ACL'25)
> [6] Information Flow Routes: Automatically Interpreting Language Models at Scale (EMNLP'24)
> [7] Circuit Stability Characterizes Language Model Generalization (ACL'25)

---

### Meta-Review · Area_Chair_YQDt · 2025-12-30

**Summary:**

This paper proposes query circuit discovery as a way to obtain local, input-level explanations for individual LLM outputs, in contrast to prior work that focuses on dataset-level or capability-level circuits. To support robustness and stability, the authors introduce the Normalized Deviation Faithfulness (NDF) metric and a Best-of-N (BoN) sampling strategy over paraphrases. Experiments across IOI, arithmetic, MMLU, and ARC show that highly sparse subnetworks can retain a substantial fraction of per-query performance. In response to reviewer feedback, the rebuttal adds scaling experiments up to 8B parameters and further analyses of circuit overlap and variance across paraphrases.

Several reviewer concerns were substantially addressed. In particular, the added large-model experiments reduce the worry that the results only hold at toy scale. New analyses of overlap and shared sub-circuits across paraphrases partially address concerns that BoN yields arbitrary subnetworks, and necessity-style ablations strengthen the claim that the identified circuits are not entirely trivial.

However, decision-critical concerns remain, leaving the paper borderline. The central unresolved issue is conceptual: showing that a sparse subgraph is sufficient or necessary for preserving performance does not by itself establish that it captures the model’s actual computation for a given query, rather than one of many alternative subnetworks capable of reproducing the output. Multiple reviewers emphasize the gap between identifying working subnetworks and delivering genuine mechanistic explanations. The submission lacks a concrete, end-to-end interpretability case study demonstrating how query circuits lead to human-meaningful insight or actionable understanding. Relatedly, the distinction between “query circuits” and task-general or capability circuits remains ambiguous, especially in settings such as IOI where overlap is substantial. Concerns about shortcut or hard-coded circuits are discussed but not decisively ruled out by the current evidence. Finally, some of the most persuasive demonstrations are deferred to a future camera-ready version, which weakens the present submission.

Overall, while the paper introduces an interesting and potentially important direction and the rebuttal resolves several empirical concerns, the remaining conceptual and interpretability gaps prevent a clear acceptance at this stage. The work would benefit from revision with stronger, concrete demonstrations of explanatory power and clearer grounding of what constitutes a true query-level mechanism.

**Reviewer Concerns:**

Concerns that were addressed by the rebuttal: The reviewers asked for scaling evidence beyond small models, and the rebuttal adds experiments on larger models (including 8B) showing broadly similar trends, reducing the concern that the findings are confined to toy-scale settings. The rebuttal also responds to requests to quantify circuit variance and overlap across paraphrases by adding analyses (e.g., similarity/overlap statistics and “shared sub-circuit” evidence), which supports the claim that BoN is not producing completely arbitrary subnetworks and that some consistent structure persists across perturbations. Finally, the authors provide additional arguments and tests aimed at non-triviality, including necessity-style ablations on complement circuits, which strengthen the case that the discovered subgraphs matter for performance.

Some concerns may still remain outstanding (and appear decision-critical), making it a borderline paper. Firstly, demonstrating that a sparse subgraph is sufficient or necessary for preserving performance does not by itself establish that it explains the model’s actual computation for a specific query, rather than representing one of many viable subnetworks that reproduce the output. Reviewers 3hrb and 1YXG continue to emphasize the gap between identifying sparse subnetworks that work and providing genuine mechanistic understanding. The rebuttal does not include a concrete, end-to-end interpretability case study showing how query circuits yield human-meaningful explanations or actionable insights. Relatedly, ambiguity remains around the definition of “query circuits” versus task-general circuitry, especially in settings like IOI where overlap with capability circuits is substantial. Concerns about potential hard-coding or shortcut circuits are discussed but not decisively ruled out by the current evaluation. Finally, key persuasive evidence, such as concrete usage examples or more interpretable demonstrations, is deferred to the camera-ready version, which weakens the strength of the present submission. I believe the authors should consider revising these issues and the paper could be published in the next venue.

**Reviewer Scores:**

Reviewer 7mAG: 6 (marginal accept). With the added scaling and variance analyses, they might move slightly upward in confidence but their main caution about brittleness and limited novelty likely remains. I would expect no change, but still in the “borderline accept” region.

Reviewer UiWe: 8 (accept). Their main request was more discussion of usage and applications; the rebuttal reiterates planned examples rather than providing them, but the added experiments support their positive view. I would expect no change (stay 8).

Reviewer 3hrb: 4 (marginal reject). They explicitly state after the rebuttal that their critical concern about the definition and explainability of “query circuits” remains, and they will maintain their score. I would expect no change (stay 4).

Reviewer 1YXG: 2 (reject). The rebuttal addresses several empirical requests (NFS on paraphrases/averaging baseline, added clarifications, additional experiments) and provides counterfactual-style necessity arguments, but their main soundness concern (faithfulness and hard-coding, missing grounding in counterfactual notions, lack of interpretability demonstrations) is only partially mitigated. I would expect a small upward adjustment at best (to 4) if they felt the new experiments sufficiently reduce uncertainty, but it is also plausible they remain at 2. My best estimate is 2 to 4, possibly.

---

### Decision · Program_Chairs · 2026-01-26

Reject